# SAGE: Shaping Anchors for Guided Exploration in RLVR of LLMs

Chanuk Lee [1]   Minki Kang [1]   Sung Ju Hwang [1 2]

## Abstract

Recent studies observe that reinforcement learning with verifiable rewards (RLVR) reliably improves pass@1 on reasoning tasks, yet often fails to yield comparable gains in pass@k, raising the question of whether RLVR genuinely enables large language models to acquire novel reasoning abilities or merely enhances the efficiency of sampling reasoning modes already present in the base model. Prior analyses largely support the latter view, attributing this limitation to structural properties of standard RLVR objectives that result in insufficient exploration pressure. In this work, we argue that a central structural constraint arises from reverse-KL regularization, which stabilizes training but inherently anchors the policy to the reference distribution, thereby suppressing the emergence of alternative reasoning modes. However, we show that neither removing the KL term nor replacing it with forward-KL provides a satisfactory solution, as both disrupt the efficiency–coverage trade-off by either inducing reward hacking or allocating probability mass to off-target regions. To resolve this tension, we propose SAGE, a principled framework that enables controllable empirical support expansion by reshaping the reverse-KL anchor distribution itself through a guide function $q(x, y)$, achieving consistent improvements in both pass@1 and pass@k across challenging mathematical reasoning benchmarks. Our code is available at https://github.com/tally0818/SAGE.

## 1. Introduction

Large language models (LLMs) have achieved substantial progress on complex reasoning tasks, driven by advances in supervised post-training and reinforcement learning-based alignment. Among these approaches, reinforcement learning with verifiable rewards (RLVR) has emerged as a practical and effective framework for improving multi-step reasoning in domains such as mathematical problem solving and code generation (Lambert et al., 2024; Shao et al., 2024; Yu et al., 2025). By exploiting verifiable reward signals, RLVR enables scalable optimization of reasoning trajectories without reliance on human preference data.

Despite these successes, recent analyses (Yue et al., 2025; Wu et al., 2025) demonstrate that RLVR predominantly reinforces a narrow subset of a model's pre-existing reasoning modes. During training, the policy frequently collapses onto a small collection of familiar solution paths, leading to support shrinkage. This mode-sharpening behavior highlights a fundamental need for effective exploration strategies in RLVR. Subsequent work (GX-Chen et al., 2025) further indicates that KL-regularized RL can be suboptimal not only in terms of exploration but also exploitation. While GX-Chen et al. (2025) propose reward shaping as a partial remedy for the exploitation bias, the design of mechanisms that enable systematic and sustained exploration in RLVR remains an open challenge.

To address this exploration challenge, a broad class of heuristic approaches has been proposed, with a dominant line of work focusing on modifying the KL regularization term. Originally introduced to constrain excessive deviation from the reference model, the KL penalty often induces distributional sharpening and suppresses low-support trajectories. Recent studies have explored eliminating the KL penalty altogether (Yu et al., 2025; Yang et al., 2025; Liu et al., 2025a), replacing reverse KL with forward-KL (Deng et al., 2025b), generalizing to alternative $f$-divergences (Li et al., 2025), or applying annealed KL schedules (Chen et al., 2025b).

In contrast, our work stems from a key insight: *both discarding the reverse-KL term and replacing it with forward-KL lead to suboptimal exploration dynamics*. The reverse-KL divergence—often viewed as suppressing exploration—serves as a critical stabilizing anchor that prevents overfitting to sparse reward signals. While substituting it with forward-KL divergence can formally enlarge the support, it frequently allocates probability mass to reward-irrelevant regions, yielding limited improvements in pass@1.

[1]KAIST [2]DeepAuto.ai. Correspondence to: Sung Ju Hwang <sjhwang@kaist.ac.kr>.

*Proceedings of the $43^{rd}$ International Conference on Machine Learning*, Seoul, South Korea. PMLR 306, 2026. Copyright 2026 by the author(s).

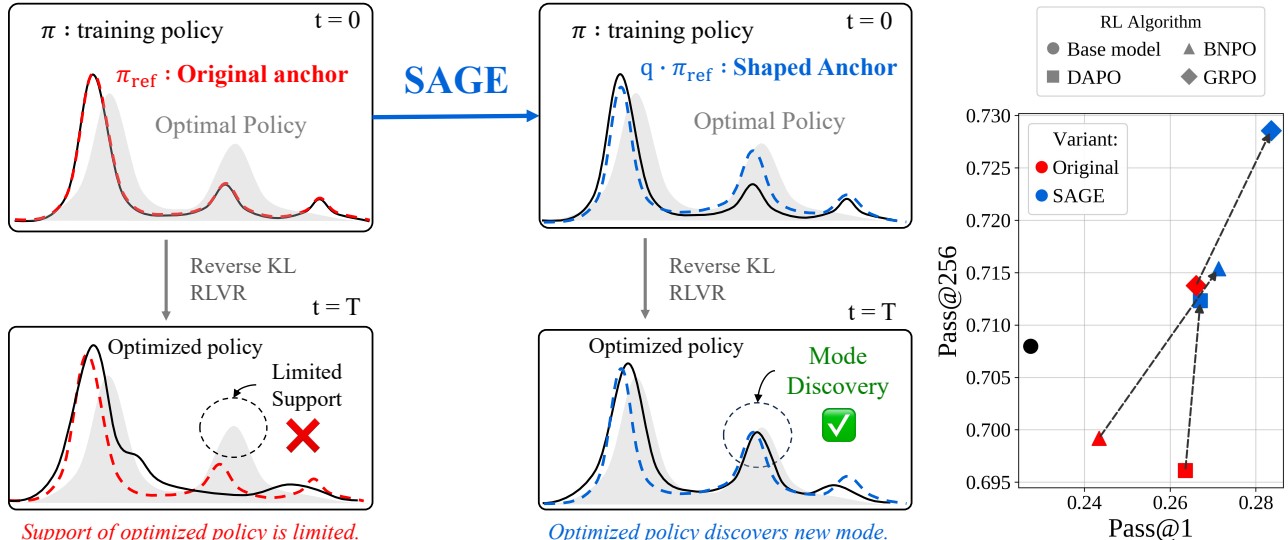

*Figure 1.* **Conceptual illustration of SAGE. Left:** Standard reverse-KL RLVR with the original anchor leads to mode collapse, failing to cover valid but low-density regions. **Middle:** SAGE introduces a multiplicative *guide function* $q$ that forms a shaped anchor $q \cdot \pi_{\text{ref}}$ to guide exploration, enabling the recovery of underexplored yet reward-compatible reasoning modes. **Right:** Performance comparison on math reasoning tasks (AIME, AMC23, MATH-500) shows that applying SAGE (blue) consistently outperform baselines in both Pass@1 and Pass@256 metrics across three different RL algorithms with Qwen-2.5-7B-Math model. See Section 5 for more details.

This leads to the following question:

*Can we retain the stabilizing benefits of reverse-KL while explicitly enabling controlled empirical support expansion?*

To address this question, we introduce Shaping Anchors for Guided Exploration (**SAGE**), a framework that constructs a *guide function* $q(x, y)$ to guide the policy toward underexplored or previously inaccessible reasoning modes by *reshaping the distribution toward which the KL term anchors the policy*. We show that $q$ can be instantiated using lightweight intrinsic signals that naturally arise during LLM reasoning, such as entropy and surprisal. Depending on how these signals are aggregated, SAGE yields several practical variants, including random exploration, token-level exploration, and branch-level exploration. Although these variants differ in implementation, they share a unifying principle: guided exploration is achieved through a modification of the KL anchor. Figure 1 provides a conceptual illustration of how these variants instantiate a shared exploration mechanism.

We evaluated multiple instantiations of SAGE on the AIME, AMC23, and MATH-500 datasets and observed consistent improvements in both pass@1 and pass@k across diverse RLVR settings, demonstrating that principled exploration is compatible with reverse-KL-based regularization.

Our main contributions are as follows:

- We analyze the limitations of existing exploration strategies in reverse-KL RLVR and show that controllable mode expansion can be achieved by *leveraging*, rather than weakening, the reverse-KL anchor.

- We propose **SAGE**, a principled framework for constructing exploration-enhanced anchor distributions that promote reasoning-mode expansion while preserving stability.

- We instantiate SAGE using intrinsic reasoning signals, such as surprisal and entropy, yielding a unified family of practical exploration mechanisms.

- We demonstrate through extensive experiments that SAGE-style variants consistently improve reasoning accuracy and exploration capability on mathematical reasoning benchmarks.

## 2. Related Works

**RLVR and Mode Collapse.** Reinforcement learning with verifiable rewards (RLVR) has emerged as an effective framework for improving multi-step reasoning in large language models, achieving strong empirical performance on tasks such as mathematical problem solving and code generation (Lambert et al., 2024; Shao et al., 2024; Yu et al., 2025). Despite these successes, a growing body of evidence indicates that RLVR predominantly induces *mode sharpening* rather than *mode discovery*. Recent analyses report rapid entropy collapse, support saturation, and even support shrinkage, suggesting that standard RLVR pipelines tend to reinforce a narrow subset of pre-existing reasoning modes (Yue et al., 2025; Wu et al., 2025). These findings highlight that the core challenge in RLVR lies in exploration.

**Weakening or Removing the KL Anchor.** A prominent line of work attempts to alleviate mode collapse by modifying the KL regularization that anchors the policy to a reference model. Representative approaches include removing the KL penalty (Yu et al., 2025; Yang et al., 2025; Liu et al., 2025a), replacing reverse-KL with forward-KL (Deng et al., 2025b), generalizing to alternative $f$-divergences (Li et al., 2025), or applying annealed KL schedules (Chen et al., 2025b). While these methods can enlarge the effective support and improve coverage-oriented metrics such as pass@k, weakening the anchoring constraint often destabilizes optimization, leading to reward hacking or degraded pass@1 performance. These trade-offs suggest that simply relaxing the KL anchor does not provide a principled solution to the exploration-exploitation dilemma in RLVR.

**Exploration via Sampling and Objective Heuristics.** Another line of work improves exploration while retaining the standard reverse-KL anchor. These approaches include modifying sampling dynamics (e.g., temperature tuning and adaptive sampling schedules) (Yu et al., 2025; Liu et al., 2025b; Chen et al., 2025b; Hou et al., 2025b), introducing entropy- or uncertainty-based regularization and advantage shaping (Chen et al., 2025a; Xie et al., 2025; Cheng et al., 2025; Deng et al., 2025a; Jiang et al., 2025; Zhang et al., 2025a), and directly optimizing pass@!$k$ objectives (Chen et al., 2025c; Walder & Karkhanis, 2025). While these methods can empirically improve diversity, they remain largely heuristic and do not fundamentally change the distribution toward which the policy is anchored. From an optimization perspective, prior work (Zhang et al., 2025b) addressed KL-divergence estimation issues arising from the off-policy gap between the reference and training policies, yet still maintained the *fixed-anchor* assumption. Consequently, exploration remains biased toward high-density regions of the reference policy, limiting the discovery of genuinely distinct reasoning modes.

**Ours: Anchor Shaping for Guided Exploration.** In contrast to previous approaches, our work reexamines the role of the reverse-KL divergence itself. Although commonly viewed as suppressing exploration, reverse KL provides a crucial stabilizing anchor that prevents overfitting to sparse reward signals and mitigates reward hacking. Rather than weakening or discarding this anchor, we propose to *reshape the distribution toward which it anchors the policy*. We introduce *SAGE*, a framework that treats the reverse-KL not as a constraint to be relaxed, but as an anchor whose target distribution can be deliberately shaped to guide exploration toward under-explored reasoning modes. This perspective enables controlled empirical support expansion while preserving the stability benefits of KL-based regularization.

Importantly, the SAGE family operates orthogonally to existing RLVR techniques—including advantage shaping, dynamic sampling, entropy-based bonuses, and pass@k optimization—by modifying the anchor distribution itself rather than altering the optimization structure, such as rewards or sampling heuristics. This modularity enables SAGE to be seamlessly integrated into diverse RLVR pipelines, establishing a general and principled framework for stable, KL-compatible empirical support expansion.

## 3. Preliminary

In this section, we formalize the problem setup, revisit the closed-form behavior of reverse-KL–regularized RLVR, and introduce the proposed *SAGE*, a general mechanism for uncovering *underexplored reasoning modes* hidden in the low-density regions of the reference policy.

### 3.1. Notations and Setup

Let $x \in \mathcal{D}$ denote an input problem and $y = (y_{1:T})$ a generated trajectory. We denote the trainable policy by $\pi_\theta(y \mid x)$ and the reference policy by $\pi_{\text{ref}}(y \mid x)$. Each trajectory receives a verifiable reward $r(x, y) \in \{0, 1\}$. We use $D_{\text{KL}}(\cdot \| \cdot)$ to denote the Kullback-Leibler divergence and let $\beta > 0$ be the reverse-KL regularization coefficient.

### 3.2. Reverse-KL RLVR and Density Collapse

Standard reverse-KL RLVR optimizes the objective

$$J(\theta) = \mathbb{E}_{x, y \sim \pi_\theta}\Big[ r(x, y) - \beta\, D_{\text{KL}}(\pi_\theta \,\|\, \pi_{\text{ref}}) \Big]. \quad (1)$$

Its stationary solution admits the exponential tilting form:

$$\pi_\theta(y \mid x) \propto \pi_{\text{ref}}(y \mid x) \exp\Big( \tfrac{1}{\beta} r(x, y) \Big). \quad (2)$$

Because the update is multiplicative with respect to the reference distribution, reverse-KL RLVR exhibits two characteristic structural behaviors:

- **Mode Sharpening.** Probability mass increasingly concentrates in the high-density regions of $\pi_{\text{ref}}$, thereby overshadowing low-density regions that may contain alternative reasoning branches.

- **Low-Density Suppression.** Even when low-density trajectories receive positive reward, the multiplicative update in Eq. (2) amplifies their probability only proportionally to their initial density. As a result, reasoning branches encoded in $\pi_{\text{ref}}$ but sampled infrequently remain effectively inaccessible.

In summary, reverse-KL RLVR achieves training stability at the cost of *limited exploration within the reference support*,

systematically neglecting rare but reward-valid trajectories that could correspond to distinct reasoning strategies. A detailed analysis is provided in Section A.

# 4. SAGE: Shaping Anchors for Guided Exploration

To mitigate the mode collapse induced by exponential tilting under reverse-KL regularization, we propose to shape the KL anchor by introducing a trajectory-dependent *guide function* $q(x, y) > 0$:

$$J'(\theta) = \mathbb{E}_{x,y\sim\pi_\theta}\left[r(x,y) - \beta\, D_{\mathrm{KL}}(\pi_\theta \,\|\, q \cdot \pi_{\mathrm{ref}})\right]. \quad (3)$$

During policy optimization, the guide function $q$ is treated as fixed with respect to $\theta$ (i.e., gradients are not propagated through $q$). In particular, although $q$ may be instantiated from policy-dependent statistics (e.g., entropy or surprisal), it is held constant during each policy update and updated only across iterations. Under this per-iteration fixed-anchor assumption, the modified objective admits the following stationary solution:

$$\pi_\theta(y \mid x) \propto q(x,y)\,\pi_{\mathrm{ref}}(y \mid x)\,\exp\!\left(\tfrac{1}{\beta}r(x,y)\right). \quad (4)$$

While the theoretical support remains unchanged-namely, for any trajectory $y$, $\pi_\theta(y \mid x) = 0$ if and only if $\pi_{\mathrm{ref}}(y \mid x) = 0$-the formulation reshapes the *empirical support* (Def. 4.1) through targeted probability redistribution. The guide function $q$ amplifies low-density yet high-potential reasoning regions, mitigating the exploration bias of reverse-KL-based RLVR while respecting the anchoring constraint of $\pi_{\mathrm{ref}}$.

Notably, the product $q \cdot \pi_{\mathrm{ref}}$ is not required to be a normalized distribution. To accommodate this, we introduce the *pseudo–KL divergence*

$$\tilde{D}_{\mathrm{KL}}(p\|f) = \mathbb{E}_{y\sim p}\left[\log\frac{p(y)}{f(y)}\right], \quad (5)$$

which is well-defined for any positive, potentially unnormalized function $f$. Crucially, replacing the standard KL divergence with its pseudo counterpart preserves the same stationary proportional solution as Eq. (4), while enabling principled policy optimization with an unnormalized anchoring distribution. The resulting objective can thus be equivalently written as

$$J'(\theta) = \mathbb{E}_{x,y\sim\pi_\theta}\left[r(x,y) - \beta\, \tilde{D}_{\mathrm{KL}}(\pi_\theta \,\|\, q \cdot \pi_{\mathrm{ref}})\right]. \quad (6)$$

Moreover, by expanding the pseudo-KL regularizer using the token-level factorization of the guide function

$$q(x,y) = \prod_{t=1}^{T} q(y_t \mid x, y_{<t}), \quad (7)$$

we obtain the following decomposition:

$$\begin{aligned}\tilde{D}_{\mathrm{KL}}(\pi_\theta \,\|\, q \cdot \pi_{\mathrm{ref}}) &= D_{\mathrm{KL}}(\pi_\theta \,\|\, \pi_{\mathrm{ref}}) \\ &- \mathbb{E}_{x,y\sim\pi_\theta}\left[\sum_{t=1}^{T}\log q(y_t \mid x, y_{<t})\right].\end{aligned} \quad (8)$$

This formulation reveals that SAGE *modifies the KL leash along the guide function* $q$, counteracting the mode-sharpening effect while preserving its stabilizing role during optimization, rather than globally smoothing the policy.

## 4.1. Empirical Support Expansion with SAGE

In this section, following the concepts from Wu et al. (2025), we analyze the empirical support properties of the proposed SAGE framework and demonstrate that it enables empirical support expansion, in contrast to standard reverse-KL-based RLVR methods.

**Definition 4.1** (Empirical Support). Let $\mathcal{C} = \{y \in \mathcal{Y} \mid r(x,y) = 1\}$ denote the set of reward-valid completions for a given input $x$, and let $\epsilon > 0$ be a probability threshold. The *empirical support* of a conditional distribution $p(y \mid x)$ is defined as

$$\mathrm{supp}_\epsilon(p) := \big\{y \in \mathcal{C} \mid p(y \mid x) > \epsilon\big\}.$$

Prior work (Wu et al., 2025) established the following negative result for reverse-KL RLVR methods.

**Theorem 4.2** (Empirical Support Preservation). *Let $\pi_{RLVR}$ denote a reverse-KL-based RLVR policy (e.g., PPO/GRPO), and let $\pi_{ref}$ be the reference model. Assume that $\epsilon$ lies below the finite-sample detectability threshold induced by the rollout budget. Then, under standard sampling and policy update procedures:*

$$\mathrm{supp}_\epsilon\big(\pi_{RLVR}(\cdot \mid x)\big) \subseteq \mathrm{supp}_\epsilon\big(\pi_{ref}(\cdot \mid x)\big).$$

Theorem 4.2 implies that conventional RLVR algorithms are fundamentally incapable of achieving empirical support expansion: any reward-valid completion assigned non-negligible probability by the trained policy must already belong to the empirical support of the reference model.

We now show that this limitation can be overcome by SAGE. Let $\pi_{\mathrm{SAGE}}$ denote the policy obtained under the SAGE framework with a guide function $q(x, y)$.

**Theorem 4.3** (Empirical Support Expansion with SAGE). *Fix an input $x$, a probability threshold $\epsilon > 0$ and $y^\star \in \mathcal{C} \setminus \mathrm{supp}_\epsilon(\pi_{ref}(\cdot|x))$. If we design the guide function $q(x, y)$ such that:*

$$q(x, y^\star) > \frac{\epsilon}{\pi_{RLVR}(y^\star \mid x)}\, \mathbb{E}_{y\sim\pi_{RLVR}}\big[q(x,y)\big],$$

*then:*

$$y^\star \in \operatorname{supp}_\epsilon\big(\pi_{SAGE}(\cdot|x)\big).$$

Theorem 4.3 establishes that, in contrast to standard reverse-KL RLVR, the SAGE framework is theoretically capable of recovering reward-valid solutions that lie outside the empirical support of the reference policy. By appropriately designing the guide function $q(x, y)$, SAGE can allocate non-negligible probability mass to such underrepresented yet valid solutions, thereby enabling explicit empirical support expansion beyond the reference model. A formal proof is included in Section A.3.

Nevertheless, Theorem 4.3 constitutes an existence result. It provides a sufficient condition under which a reward-valid but underrepresented solution $y^\star$ enters the empirical support of $\pi_{SAGE}$, but does not prescribe how to realize such a guide function in practice. In particular, during training, the probability of reward-valid solutions under $\pi_{RLVR}$ is not directly observable, which precludes the use of explicit reward information in constructing $q(x, y)$. This gap between theoretical sufficiency and practical implementability naturally leads to the following question:

*How should we design the guide function $q(x, y)$ to reliably highlight underexplored reasoning modes?*

To address this challenge, we propose to construct the guide function using reward-agnostic intrinsic signals that correlate with underexplored reasoning modes. In the next section, we identify such intrinsic exploration proxies within LLM reasoning trajectories and demonstrate how they can be systematically leveraged to instantiate effective guide functions within the SAGE framework.

### 4.2. Intrinsic Signals for Identifying Underexplored Regions

To construct useful forms of $q(x, y)$, we analyze intrinsic signals in LLM reasoning trajectories. Consistent with prior work, we find:

- **Surprisal** ($-\log p$) predominantly reacts to the rarity of lexical forms—e.g., uncommon numerical expressions or rare phrasing—and thus tends to capture surface-level deviations rather than structural reasoning differences. (Xie et al., 2025)

- **Entropy** ($-\sum p \log p$) reflects the size of the local decision set. High-entropy regions correspond to branching decisions where multiple solution paths are viable, whereas entropy sharply decreases in deterministic symbolic calculations. (Cheng et al., 2025; Jiang et al., 2025)

These results indicate that entropy provides a more structurally meaningful signal for detecting reasoning branches, whereas surprisal captures lexical outliers.

### 4.3. Designing the Guide Function $q$

The decomposition in Eq. (8) shows that $q(x, y)$ governs how aggressively low-density trajectories are revisited. We instantiate representative families covering different granularities of exploration.

#### 4.3.1. RANDOM EXPLORATION

As a minimal baseline, we consider a random-search-based guide function:

$$q_{\text{Random}}(y_t \mid x, y_{<t}) = \begin{cases} 1 & \text{with probability } \epsilon, \\ \mathcal{N}(1, \sigma^2) & \text{otherwise,} \end{cases} \tag{9}$$

where $\epsilon$ and $\sigma$ are controlled by cosine-decay schedules.

Under this formulation, Eq. (6) induces a stochastic local perturbation around the reference distribution. While this strategy encourages exploration in a generic sense, it remains agnostic to problem structure, uncertainty, or mode diversity, and therefore provides limited capability for targeted mode discovery. This limitation motivates the use of intrinsic signals to guide exploration, leading to the following variants.

#### 4.3.2. TOKEN-LEVEL EXPLORATION

To encourage uncommon lexical choices, we define a surprisal-aware variant:

$$q_{\text{Token}}(y_t \mid x, y_{<t}) = \mathcal{N}(1 + \alpha w_t, w_t^2 \sigma^2), \tag{10}$$

where $w_t = \text{MiniMaxScaler}(-\log \pi_\theta(y_t \mid x, y_{<t}))$, and $\alpha$ and $\sigma$ follow cosine-decay schedules.

This modulation biases exploration toward low-probability tokens, improving lexical novelty. However, surprisal is primarily a *token-level* rarity signal: it can be high even when the underlying reasoning trajectory is unchanged (e.g., stylistic variation or paraphrasing), and thus provides a weak proxy for branching in reasoning space.

As discussed in Section 4.2, we instead use entropy as an intrinsic signal. Entropy is not a perfect *reasoning-branch* indicator either: it often spikes at *narrative branch points* such as problem restatement, alternative formulations, or high-level strategy framing, where multiple plausible continuations exist. Nevertheless, among lightweight token-level signals, entropy remains the relatively effective for promoting diverse continuations that frequently coincide with underexplored reasoning branches. We therefore adopt entropy-based modulation in the next section.

*Table 1.* **Main results of the GRPO family on AIME, AMC23, and MATH-500.** We report pass@1 and pass@256 (mean $\pm$ std over 5 seeds). **Bold** numbers indicate the best performance for each dataset and metric. GRPO+BRANCH achieves the strongest results on average across all datasets. The Avg column reports the average performance over the three benchmarks.

| Algorithm | AIME | | AMC23 | | MATH-500 | | Avg | |
|---|---|---|---|---|---|---|---|---|
| | Pass@1 | Pass@256 | Pass@1 | Pass@256 | Pass@1 | Pass@256 | Pass@1 | Pass@256 |
| Base Model | 0.033±0.024 | 0.393±0.025 | 0.355±0.062 | 0.950±0.018 | 0.294±0.018 | 0.781±0.009 | 0.227 | 0.708 |
| GRPO | 0.053±0.027 | 0.407±0.025 | 0.430±0.057 | 0.960±0.014 | 0.315±0.036 | 0.775±0.020 | 0.266 | 0.714 |
| GRPO w/o KL | 0.043±0.019 | 0.410±0.025 | 0.410±0.034 | 0.930±0.033 | 0.319±0.019 | 0.763±0.010 | 0.258 | 0.701 |
| GRPO + Forward KL | 0.057± 0.035 | **0.433±0.012** | 0.420±0.021 | **0.975±0.000** | 0.318±0.034 | 0.772±0.007 | 0.265 | 0.727 |
| *SAGE (Ours)* | | | | | | | | |
| GRPO + Random | 0.050±0.041 | 0.420±0.030 | 0.410±0.049 | 0.965±0.029 | **0.340±0.026** | 0.782±0.014 | 0.267 | 0.722 |
| GRPO + Token | 0.057±0.019 | 0.427±0.025 | 0.395±0.037 | 0.960±0.014 | 0.303±0.042 | 0.769±0.005 | 0.252 | 0.718 |
| GRPO + Branch | **0.063±0.030** | 0.427±0.035 | **0.465±0.063** | 0.965±0.029 | 0.322±0.037 | **0.794±0.011** | **0.284** | **0.729** |

### 4.3.3. BRANCH-LEVEL EXPLORATION

Motivated by the use of entropy as an indicator of reasoning branch proliferation in prior works (Cheng et al., 2025; Hou et al., 2025a; Wang et al., 2026), and inspired by dynamic range compression mechanisms in audio signal processing, we design the following threshold–ratio rule:

$$q_{\text{Branch}}(y_t \mid x, y_{<t}) = 1 + \gamma \left(\mathcal{H}_t - \tau\right)^+, \quad (11)$$

where $\mathcal{H}_t$ denotes token-level entropy:

$$\mathcal{H}_t = -\sum_{v \in \mathcal{V}} \pi_\theta(v \mid x, y_{<t}) \log \pi_\theta(v \mid x, y_{<t}), \quad (12)$$

and $\gamma$ and $\tau$ respectively control the magnitude and activation threshold of exploration.

This formulation selectively amplifies exploration only at states exhibiting substantial uncertainty, while suppressing unnecessary perturbations in low-entropy regions.

A toy example illustrating that the proposed design can satisfy Theorem 4.3 is presented in Section C. Hyperparameter ablation studies are reported in Section D.

Notably, these intrinsic signals are obtained on-the-fly from the reasoning process itself, allowing $q(x, y)$ to be instantiated without extra forward passes, and thus maintaining computational efficiency.

## 5. Experiments

### 5.1. Experimental Setup

**Model.** We use `Qwen2.5-Math-7B-Base` (Yang et al., 2024) as the backbone model considering their wide adoption in previous works on RLVR (Xiao et al., 2025; Li et al., 2025; Chen et al., 2025a; Cheng et al., 2025; Deng et al., 2025a). We also use `DeepSeek-R1-Distill-Qwen-7B` (DeepSeek-AI) to show our method is applicable to long reasoning model.

**Training Procedure.** We adopt a two-stage training pipeline. We first perform a light supervised fine-tuning (SFT) on a small subset of numerically answerable instances from `OpenMathReasoning-mini` (Daniel Han & team, 2024) to standardize the model's output format. The resulting checkpoint is used as the baseline model for subsequent RLVR. We then conduct RLVR on `DAPO-Math-17K-Processed` (Yu et al., 2025) using `Math-Verify` (HuggingFace, 2024) as the reward function, with LoRA (Hu et al., 2022) adapters applied throughout.

**Baselines & Ours.** In the following experiments, (1) Baseline denotes the supervised fine-tuned (SFT-only) model. (2) GRPO refers to the model trained with GRPO (Shao et al., 2024), while (3) GRPO w/o KL denotes the GRPO variant trained without the KL regularization term. Accordingly, (4) GRPO + Forward KL indicates the model trained with GRPO where the standard reverse-KL regularization is replaced by the forward-KL divergence. Finally, Ours (+ RANDOM, + TOKEN, and + BRANCH) refers to the SAGE-augmented variants equipped with the guide functions defined in Eq. (9), Eq. (10), and Eq. (11), respectively.

**Evaluation Datasets & Metrics.** We evaluate on three math-reasoning datasets: AIME 2024-2025 (60 problems) (Mathematical Association of America, 2025), AMC23 (40 problems) (Mathematical Association of America, 2023), and the level-5 subset of MATH-500 (134 problems) (Hendrycks et al., 2021). Outputs are graded using `Math-Verify` (HuggingFace, 2024), and we report pass@1 and pass@256.

**Additional Details.** Full training details are given in Section E.

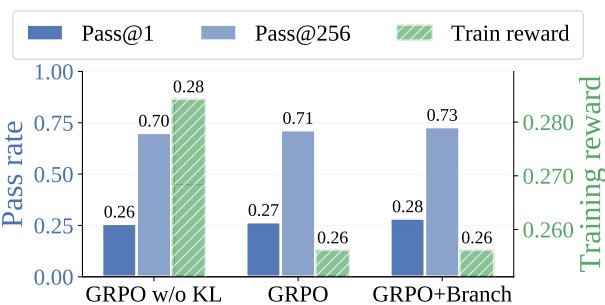

*Figure 2.* **Comparison between GRPO without KL regularization, GRPO, and GRPO + Branch.** We report dataset-averaged pass@1, pass@256, and the average training reward (over the last 30 RL steps). We observed that discarding KL regularization successfully induced higher training reward but failed to yield higher pass rate both on pass@1 and pass@256.

## 5.2. Main Results

Table 1 summarizes our main experimental results. Overall, incorporating SAGE into GRPO leads to consistent performance improvements when the guide function is appropriately structured. Among the examined variants, GRPO+BRANCH achieves the strongest average gains, improving both dataset-averaged pass@1 (0.284) and pass@256 (0.729) compared to vanilla GRPO. At the dataset level, it yields the highest pass@1 on AIME and AMC23, as well as the best pass@256 on MATH-500, demonstrating robust benefits across diverse reasoning benchmarks.

In contrast, GRPO+RANDOM, TOKEN exhibits less consistent behavior, occasionally improving specific metrics but failing to deliver stable gains across datasets. These observations suggest that the effectiveness of SAGE critically depends on exploration strategies, with structured branch-level guidance serving as a particularly effective instantiation. To better understand the underlying mechanism,

Figure 2 analyzes the role of reverse-KL regularization. Although removing the reverse-KL penalty leads to substantially higher training rewards, it results in worse test-time performance, indicating severe reward overfitting. This suggests that reverse-KL regularization plays a critical role in preventing policy collapse and stabilizing learning. A training-dynamics interpretation of this behavior is provided in Section B.

We further compare different divergence constraints in Table 1. While forward-KL regularization improves pass@256 through broader mode coverage, its gains in pass@1 are limited. In contrast, structured exploration within SAGE achieves competitive high-sample performance while providing more reliable gains in low-sample regimes. Theoretical explanation is further elaborated in the Section A.4.

*Table 2.* **Frequency analysis of reasoning patterns.** We report the average frequency over 3 seeds of detected reasoning patterns across baseline, GRPO, and GRPO+BRANCH on the AIME dataset. Constraint setup and structural reasoning are amplified by RL training, while GRPO+BRANCH induces more *proof-by-contradiction*. Best results are in **bold**.

| Reasoning Pattern | Base Model | GRPO | GRPO + Branch |
|---|---|---|---|
| Constraint Setup | 0.458 | **0.604** | 0.583 |
| Structural Reasoning | 0.291 | **0.377** | 0.301 |
| Proof by Contradiction | 0.007 | 0.008 | **0.033** |

## 5.3. Reasoning Pattern Analysis

We analyze how different training strategies shape model reasoning behaviors by measuring the frequency of several representative reasoning patterns, including *constraint setup*, *structural reasoning*, and *proof by contradiction*. Each pattern is identified using lightweight keyword-based heuristic detectors and evaluated only on solutions that produce correct final answers.

The frequency of a pattern $X$ is computed as:

$$\text{freq}(X) = \frac{1}{N} \sum_{i=1}^{N} \left( \frac{1}{C_i} \sum_{j=1}^{C_i} \mathbb{I}\{X \text{ appears in solution } j\} \right), \tag{13}$$

where $N$ denotes the number of problems and $C_i$ is the number of correct solutions for problem $i$.

Table 2 reports the resulting frequencies across baseline, GRPO, and GRPO+BRANCH. Reinforcement learning substantially increases dominant structured reasoning behaviors such as explicit constraint formulation and structural decomposition. Notably, while *proof by contradiction* remains extremely rare for both the baseline and GRPO, its occurrence increases markedly under GRPO+BRANCH, indicating that branch-level exploration recovers valid but low-probability reasoning trajectories that standard RLVR fails to discover.

More detailed quantitative analysis is provided in Section G, along with qualitative case studies in Section H.

## 5.4. Algorithm Ablations

We evaluate the compatibility of SAGE with diverse RLVR algorithms beyond GRPO, including DAPO (Yu et al., 2025) and BNPO (Xiao et al., 2025)

Table 3 demonstrates that both BNPO and DAPO exhibit a consistent trade-off under naive RL training, where pass@1 improves while pass@256 degrades relative to the baseline, indicating mode collapse. When augmented with SAGE, both variants achieve simultaneous improvements in accuracy and coverage. In particular, BNPO+BRANCH yields a substantial gain in pass@1 while surpassing the baseline pass@256, achieving the strongest overall performance

*Table 3.* **Algorithm ablations across heterogeneous RLVR variants.** We study the compatibility of SAGE with BNPO (advantage shaping) and DAPO (dynamic sampling) beyond GRPO. Under naive training, both BNPO and DAPO exhibit a consistent trade-off, improving pass@1 while degrading pass@256 relative to the baseline, indicating mode collapse. Augmenting each method with branch-level SAGE simultaneously improves accuracy and coverage across all benchmarks. We report pass@1 and pass@256 (mean ± std over 5 seeds). Best results are highlighted in **bold**.

| Algorithm | AIME | | AMC23 | | MATH-500 | | Avg | |
|---|---|---|---|---|---|---|---|---|
| | Pass@1 | Pass@256 | Pass@1 | Pass@256 | Pass@1 | Pass@256 | Pass@1 | Pass@256 |
| Base Model | 0.033±0.024 | 0.393±0.025 | 0.355±0.062 | 0.950±0.018 | 0.294±0.018 | 0.781±0.009 | 0.227 | 0.708 |
| **BNPO Variation** | | | | | | | | |
| BNPO (no KL) | 0.043±0.019 | 0.400±0.031 | 0.390±0.034 | 0.920±0.021 | 0.297±0.043 | **0.778±0.010** | 0.243 | 0.699 |
| BNPO + Branch | **0.070±0.018** | **0.420±0.007** | **0.435±0.074** | **0.950±0.018** | **0.309±0.034** | 0.776±0.018 | **0.271** | **0.715** |
| **DAPO Variation** | | | | | | | | |
| DAPO (no KL) | **0.043±0.022** | 0.387±0.014 | 0.425±0.043 | 0.930±0.021 | **0.322±0.021** | 0.772±0.010 | 0.264 | 0.696 |
| DAPO + Branch | **0.043±0.015** | **0.413±0.032** | **0.440±0.042** | **0.940±0.014** | 0.318±0.019 | **0.784±0.009** | **0.267** | **0.712** |

*Table 4.* **Model ablation with DeepSeek-R1-Distill-Qwen-7B.** We evaluate the performance of GRPO and GRPO+Branch against the basemodel on AIME, AMC23, and MATH-500. Results are reported as pass@1 / pass@64 (mean ± std over 5 seeds), with **Avg** denoting dataset-averaged performance. While GRPO improves pass@1 on most datasets, GRPO+Branch yields the strongest and most consistent gains, particularly in the high-sample regime (pass@64). Notably, augmenting GRPO with SAGE improves pass@1 while incurring a smaller degradation in pass@64 compared to naive GRPO, even for long-reasoning models. Best results are highlighted in **bold**.

| Algorithm | AIME | | AMC23 | | MATH-500 | | Avg | |
|---|---|---|---|---|---|---|---|---|
| | Pass@1 | Pass@64 | Pass@1 | Pass@64 | Pass@1 | Pass@64 | Pass@1 | Pass@64 |
| Base Model | 0.230±0.036 | 0.553± 0.007 | 0.595±0.033 | **0.955± 0.011** | 0.461±0.023 | **0.827± 0.006** | 0.429 | **0.778** |
| GRPO | 0.250±0.053 | 0.550± 0.037 | **0.635±0.045** | 0.940± 0.029 | 0.461±0.024 | 0.824± 0.007 | 0.449 | 0.771 |
| GRPO + Branch | **0.260± 0.028** | **0.567±0.039** | **0.635± 0.074** | 0.935±0.014 | **0.461± 0.012** | 0.821±0.016 | **0.452** | 0.774 |

within the BNPO family. Similarly, DAPO+BRANCH restores and exceeds the baseline pass@256 while maintaining improved pass@1, effectively eliminating the exploration–exploitation imbalance induced by naive dynamic sampling. These results suggest that SAGE serves as a general and plug-and-play mechanism for stabilizing exploration across heterogeneous RLVR frameworks.

### 5.5. Model Ablations

We evaluate SAGE on the long-reasoning model DeepSeek-R1-Distill-Qwen-7B (DeepSeek-AI) with maximum sequence length of 8,192 tokens and RL-only training.

Table 4 shows that while GRPO improves pass@1 but degrades pass@64, GRPO+BRANCH further increases Avg pass@1 (0.452) and mitigates coverage loss (pass@64: 0.774). Notably, it achieves the best performance on AIME and remains competitive across AMC23 and MATH-500.

Overall, these results demonstrate that SAGE generalizes beyond base models to long-reasoning distilled architectures, consistently improving the accuracy–coverage trade-off. This confirms that SAGE serves as a robust, model-agnostic modular augmentation for RLVR, remaining ef-

fective across fundamentally different generation dynamics and reasoning paradigms.

### 5.6. Full pass@k curves

To further analyze the accuracy-coverage tradeoff of SAGE, we report the full pass@k curves of GRPO-trained and GRPO+Branch-trained Qwen2.5-Math-7B-Base models across all three datasets using the first-seed results. All pass@k values are estimated using the unbiased estimator from Chen (2021). As shown in Figure 3, the GRPO+Branch-trained model consistently achieves a better accuracy-coverage tradeoff than the standard GRPO-trained model.

### 5.7. Out-of-Distribution Evaluation

To further evaluate the mode-collapse resistance of SAGE, we evaluate both the naive GRPO-trained and GRPO+Branch-trained Qwen2.5-Math-7B-Base models on the Knights and Knaves logical puzzle benchmark (Xie et al., 2024), which introduces out-of-distribution questions not seen in the training data.

As shown in Figure 4, SAGE remains substantially more robust under distribution shift, maintaining strong perfor-

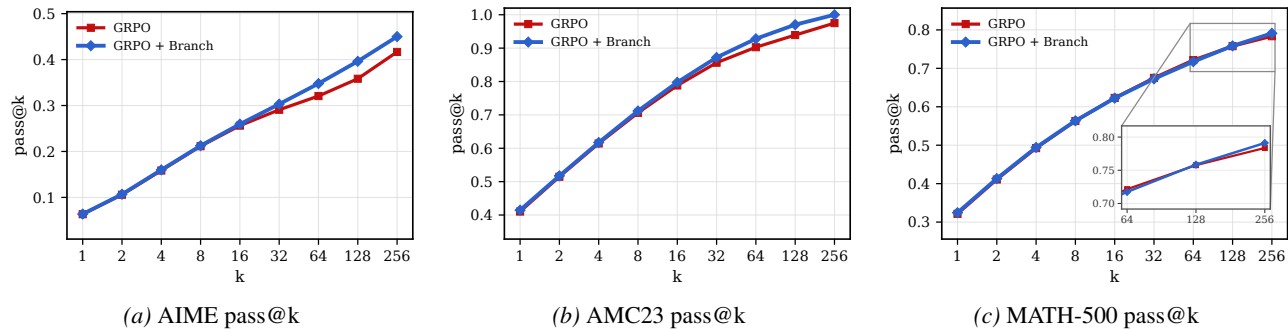

*(a) AIME pass@k*      *(b) AMC23 pass@k*      *(c) MATH-500 pass@k*

*Figure 3.* **Full pass@k curves on `Qwen2.5-Math-7B-Base`.** We compare GRPO-trained and GRPO+Branch-trained models across AIME, AMC23, and MATH-500 using the first-seed results. All pass@k values are estimated using the unbiased estimator from Chen (2021). Across all three benchmarks, SAGE consistently achieves a better accuracy-coverage trade-off than standard GRPO, improving low-$k$ accuracy while preserving broader solution coverage. As a result, SAGE maintains stronger performance also at larger $k$.

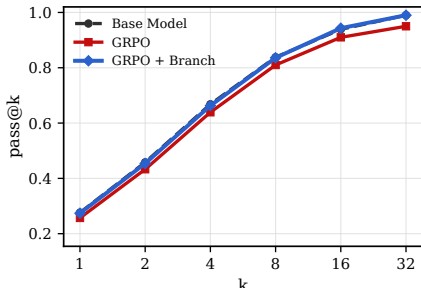

*Figure 4.* **Evaluation results on Knights and Knaves.** We report the full pass@k curves of GRPO-trained and GRPO+Branch-trained `Qwen2.5-Math-7B-Base` models on the Knights and Knaves benchmark (Xie et al., 2024). While the standard GRPO-trained model exhibits noticeable degradation under distribution shift, SAGE consistently maintains stronger performance across the entire pass@k curve, demonstrating improved robustness and resistance to mode collapse.

mance across the entire pass@k curve, whereas the standard GRPO-trained model exhibits consistent degradation. These results suggest that SAGE mitigates mode collapse and improves robustness to out-of-distribution reasoning tasks.

**Additional Experiments.** Additional experiments including evaluations on extra benchmarks and full fine-tuning on smaller models are provided in Section I. We also provide a perplexity analysis following Yue et al. (2025) in Section J.

## 6. Conclusion

We revisited reverse-KL–regularized RLVR and highlighted a fundamental trade-off: while reverse-KL stabilizes training by anchoring policies to a reference model, it also sharpens modes and suppresses diverse reasoning trajectories, whereas removing the KL term risks reward hacking. Beyond viewing reverse-KL as a mere stabilizing constraint, we reinterpret it as a controllable *exploration instrument* that shapes the empirical support of the learned policy. To overcome this limitation, we proposed SAGE, a general

framework that preserves stability while enabling principled mode expansion by augmenting the reference distribution with a guide function $q(x, y)$. This yields a tractable pseudo-KL objective that can be seamlessly integrated into existing PPO-style RLVR algorithms without altering their optimization structure. Empirically, lightweight instantiations of SAGE consistently improve both pass@1 and pass@$k$ performance across multiple reasoning benchmarks. Overall, SAGE offers a practical approach to mitigating mode sharpening in reverse-KL RLVR, and we view the systematic design or learning of more expressive guide functions as a promising direction for future work.

## 7. Limitations and Future Work

While SAGE consistently improves the accuracy-coverage trade-off across diverse datasets and algorithms, it operates within the support of a fixed reference policy under PPO-based RLVR. Specifically, SAGE redistributes probability mass among trajectories that already receive non-zero likelihood under the reference model, rather than introducing entirely new support. Thus, trajectories assigned zero probability by the reference distribution remain unreachable. This reflects an intentional scope decision rather than a limitation of the framework, as prior work has shown that RLVR is fundamentally bounded by the reasoning capacity of the base model, whereas distillation from stronger teacher models can introduce genuinely new reasoning patterns and expand the reasoning boundary (Yue et al., 2025). In this sense, distillation-based approaches and SAGE are complementary: the former expands support, while the latter enables stable and controllable exploration within it.

In addition, although SAGE is theoretically agnostic to the choice of guide function $q_\theta(x, y)$, we instantiate it using lightweight intrinsic signals such as entropy and surprisal for stability and efficiency. Learning richer guide functions from stronger teacher models or domain knowledge remains an important direction for future work.

## Acknowledgements

This work was supported by Institute for Information & communications Technology Planning & Evaluation (IITP) grant funded by the Korea government (MSIT) (RS-2019-II190075, Artificial Intelligence Graduate School Program (KAIST), No.RS-2022-II220713, Meta-learning Applicable to Real-world Problems) and the Ministry of Science and ICT (MSIT) of the Republic of Korea in connection with the Global AI Frontier Lab International Collaborative Research (No. RS-2024-00469482 & RS-2024-00509279), and National Research Foundation of Korea (NRF) grant funded by the Korea government (MSIT) (No. RS-2023-00256259), and the "Advanced GPU Utilization Support Program" funded by the Government of the Republic of Korea (Ministry of Science and ICT).

## Impact Statement

This paper presents work whose goal is to advance the field of Large Language Models, specifically focusing on exploration mechanisms in Reinforcement Learning. There are many potential societal consequences of our work, none which we feel must be specifically highlighted here.

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

# A. Theoretical analysis

The following derivations(A.1, A.2) are adapted from the closed-form analysis originally presented in Deng et al. (2025b). For completeness, we restate them using our notation and expand several intermediate steps.

## A.1. Proof of Eq. (2)

Consider the objective of maximizing

$$
\begin{aligned}
\mathcal{L}(\pi_\theta \mid x) &= \sum_{y \in \mathcal{Y}} \pi_\theta(y \mid x)\, r(x, y) \;-\; \beta \sum_{y \in \mathcal{Y}} \pi_\theta(y \mid x) \log \frac{\pi_\theta(y \mid x)}{\pi_{\mathrm{ref}}(y \mid x)} \\
&= \sum_{y \in \mathcal{Y}} \pi_\theta(y \mid x)\, r(x, y) \;-\; \beta \sum_{y \in \mathcal{Y}} \pi_\theta(y \mid x) \log \pi_\theta(y \mid x) \;+\; \beta \sum_{y \in \mathcal{Y}} \pi_\theta(y \mid x) \log \pi_{\mathrm{ref}}(y \mid x).
\end{aligned}
\tag{14}
$$

Introducing the Lagrange multiplier $\lambda$ for the normalization constraint $\sum_y \pi_\theta(y \mid x) = 1$, we obtain the augmented functional:

$$
\begin{aligned}
\tilde{\mathcal{L}}(\pi_\theta, \lambda \mid x) = &\sum_{y \in \mathcal{Y}} \pi_\theta(y \mid x)\, r(x, y) \;-\; \beta \sum_{y \in \mathcal{Y}} \pi_\theta(y \mid x) \log \pi_\theta(y \mid x) \\
&+ \beta \sum_{y \in \mathcal{Y}} \pi_\theta(y \mid x) \log \pi_{\mathrm{ref}}(y \mid x) \;+\; \lambda \left( \sum_{y \in \mathcal{Y}} \pi_\theta(y \mid x) \;-\; 1 \right).
\end{aligned}
\tag{15}
$$

The stationary condition w.r.t. $\pi_\theta$ is

$$
\frac{\partial \tilde{\mathcal{L}}}{\partial \pi_\theta} = r(x, y) - \beta(1 + \log \pi_\theta(y \mid x)) + \beta \log \pi_{\mathrm{ref}}(y \mid x) + \lambda = 0.
\tag{16}
$$

Solving for $\pi_\theta(y \mid x)$:

$$
\pi_\theta(y \mid x) = \pi_{\mathrm{ref}}(y \mid x) \exp\!\left( \tfrac{1}{\beta} r(x, y) \right) \cdot \exp\!\left( -1 + \tfrac{\lambda}{\beta} \right).
\tag{17}
$$

Absorbing the normalization term into the proportionality constant yields:

$$
\pi_\theta(y \mid x) \propto \pi_{\mathrm{ref}}(y \mid x)\, \exp\!\left( \tfrac{1}{\beta} r(x, y) \right).
\tag{18}
$$

A key observation is that this proportionality does not rely on the normalization of $\pi_{\mathrm{ref}}$. This justifies the use of the pseudo-KL divergence introduced in Eq. (5). Furthermore, while entropy regularization may increase the probability of some low-density trajectories through global smoothing, it does not enable targeted or controllable empirical support expansion, as shown below.

## A.2. Limitations of Entropy Regularization for Guided Exploration

Adding an entropy bonus $\alpha\, \mathcal{H}(\pi_\theta)$ leads to the modified objective:

$$
\begin{aligned}
\mathcal{L}(\pi_\theta \mid x) = &\sum_{y \in \mathcal{Y}} \pi_\theta(y \mid x)\, r(x, y) \\
&- (\alpha + \beta) \sum_{y \in \mathcal{Y}} \pi_\theta(y \mid x) \log \pi_\theta(y \mid x) \;+\; \beta \sum_{y \in \mathcal{Y}} \pi_\theta(y \mid x) \log \pi_{\mathrm{ref}}(y \mid x).
\end{aligned}
\tag{19}
$$

The corresponding Lagrangian becomes:

$$
\begin{aligned}
\tilde{\mathcal{L}}(\pi_\theta, \lambda \mid x) = &\sum_{y \in \mathcal{Y}} \pi_\theta(y \mid x)\, r(x, y) \;-\; (\alpha + \beta) \sum_{y \in \mathcal{Y}} \pi_\theta(y \mid x) \log \pi_\theta(y \mid x) \\
&+ \beta \sum_{y \in \mathcal{Y}} \pi_\theta(y \mid x) \log \pi_{\mathrm{ref}}(y \mid x) \;+\; \lambda \left( \sum_{y \in \mathcal{Y}} \pi_\theta(y \mid x) \;-\; 1 \right).
\end{aligned}
\tag{20}
$$

Solving the stationary condition yields:

$$\pi_\theta(y \mid x) \propto \pi_{\text{ref}}(y \mid x)^{\frac{\beta}{\alpha+\beta}} \exp\left(\frac{r(x,y)}{\alpha+\beta}\right). \tag{21}$$

This solution reveals that entropy regularization uniformly rescales the sharpness of the policy by adjusting the effective temperature, without introducing any directional preference over trajectories. Consequently, entropy bonuses act as a form of global smoothing, rather than a mechanism for selectively guiding exploration toward specific regions of the output space.

### A.3. Proof of Theorem 4.3

By Eq. (2), the RLVR policy admits the following exponential tilting form:

$$\pi_{\text{RLVR}}(y \mid x) = \frac{\pi_{\text{ref}}(y \mid x) \exp\left(\frac{1}{\beta} r(x,y)\right)}{Z(x)}, \qquad Z(x) = \sum_{y' \in \mathcal{Y}} \pi_{\text{ref}}(y' \mid x) \exp\left(\frac{1}{\beta} r(x,y')\right). \tag{22}$$

Combining Eq. (22) with Eq. (4), we obtain the following identity:

$$\pi_{\text{SAGE}}(y \mid x) = \frac{Z(x)}{Z_q(x)} \pi_{\text{RLVR}}(y \mid x) \, q(x,y). \tag{23}$$

Since

$$Z_q(x) = Z(x) \sum_{y' \in \mathcal{Y}} \pi_{\text{RLVR}}(y' \mid x) q(x,y') = Z(x) \, \mathbb{E}_{y' \sim \pi_{\text{RLVR}}(\cdot \mid x)}[q(x,y')],$$

Eq. (23) can be equivalently rewritten as

$$\pi_{\text{SAGE}}(y \mid x) = \frac{\pi_{\text{RLVR}}(y \mid x) \, q(x,y)}{\mathbb{E}_{y' \sim \pi_{\text{RLVR}}(\cdot \mid x)}[q(x,y')]}. \tag{24}$$

Now suppose that there exists guide function $q(x,y)$ such that:

$$q(x,y^\star) > \frac{\epsilon}{\pi_{\text{RLVR}}(y^\star \mid x)} \mathbb{E}_{y \sim \pi_{\text{RLVR}}(\cdot \mid x)}[q(x,y)].$$

Then it follows immediately that

$$\pi_{\text{SAGE}}(y^\star \mid x) > \epsilon, \tag{25}$$

which implies $y^\star \in \text{supp}_\epsilon(\pi_{\text{SAGE}})$, thereby completing the proof.

Section A.2 and Theorem 4.3 together highlight that SAGE enables *controllable exploration* through explicitly designed guide functions that target desired solution modes, whereas a standard entropy bonus only provides undirected, global dispersion and cannot selectively promote specific low-density solutions.

### A.4. Reducing Off-Target Probability Mass Relative to Forward-KL

Recall that the SAGE policy is defined as

$$\pi_{\text{SAGE}}(y \mid x) \propto \pi_{\text{ref}}(y \mid x) \exp\left(\frac{1}{\beta} r(x,y)\right) q(x,y), \tag{26}$$

where $q(x,y) > 0$ is a user-designed guide function.

For notational convenience, define

$$\mu(y \mid x) = \pi_{\text{ref}}(y \mid x) \exp\left(\frac{1}{\beta} r(x,y)\right).$$

Then $\pi_{\text{SAGE}}(y \mid x) = \mu(y \mid x) q(x,y) / Z_q(x)$ with $Z_q(x) = \sum_{y'} \mu(y' \mid x) q(x,y')$.

On the other hand, the stationary solution of forward-KL RLVR is given by

$$\pi_{\text{FKL}}(y \mid x) = \frac{\beta \, \pi_{\text{ref}}(y \mid x)}{\lambda - r(x,y)}, \tag{27}$$

when $\lambda (> \max r(x,y))$ is a deterministic constant induced by the normalization condition.

**Reweighting view.** Assuming $\pi_{\text{FKL}}(y \mid x) > 0$ whenever $\mu(y \mid x) > 0$, we may rewrite $\pi_{\text{SAGE}}$ as a reweighting of $\pi_{\text{FKL}}$:

$$\pi_{\text{SAGE}}(y \mid x) = \frac{\pi_{\text{FKL}}(y \mid x)\, w(x, y)}{\mathbb{E}_{y' \sim \pi_{\text{FKL}}(\cdot \mid x)}[w(x, y')]}, \tag{28}$$

where the reweighting factor is

$$w(x, y) = \frac{\mu(y \mid x)\, q(x, y)}{\pi_{\text{FKL}}(y \mid x)} = \frac{\lambda - r(x, y)}{\beta} \exp\left(\tfrac{1}{\beta} r(x, y)\right) q(x, y). \tag{29}$$

Notably, the reference policy $\pi_{\text{ref}}$ cancels out, and the relative behavior of $\pi_{\text{SAGE}}$ and $\pi_{\text{FKL}}$ is fully determined by the reward $r(x, y)$ and the guide function $q(x, y)$.

**Selective reduction of off-target mass.** Let $\mathcal{C} \subseteq \mathcal{Y}$ denote the set of reward-valid solutions(Def.4.1). If the reweighting factors satisfy the separation condition

$$\min_{y \in \mathcal{C}} w(x, y) \; > \; \max_{y \notin \mathcal{C}} w(x, y), \tag{30}$$

and $\pi_{\text{FKL}}(\mathcal{C} \mid x) > 0$, then

$$\sum_{y \notin \mathcal{C}} \pi_{\text{SAGE}}(y \mid x) < \sum_{y \notin \mathcal{C}} \pi_{\text{FKL}}(y \mid x). \tag{31}$$

That is, SAGE assigns strictly less probability mass to reward-invalid regions than the forward-KL stationary solution.

**A sufficient condition on $q$.** Since $r(x, y) = 1$ for all $y \in \mathcal{C}$ and $r(x, y) = 0$ for all $y \notin \mathcal{C}$. Then a sufficient condition for (30) is

$$\min_{y \in \mathcal{C}} q(x, y) \; > \; \frac{\lambda}{(\lambda - 1) \exp(1/\beta)} \; \max_{y \notin \mathcal{C}} q(x, y). \tag{32}$$

Following from $\beta \ll 1$, the exponential factor $\exp(-\tfrac{1}{\beta})$ is typically small, implying that even a weak preference of $q$ toward $\mathcal{C}$ is sufficient.

**Implication.** While forward-KL encourages broad mode coverage, it does so indiscriminately, potentially allocating probability mass to many reward-irrelevant regions. In contrast, SAGE introduces a controllable multiplicative factor $q(x, y)$ that can selectively amplify reward-valid modes, thereby reducing off-target probability mass without sacrificing coverage.

**Visualization.** For intuitive understanding, we provide a visualization illustrating the mode-seeking and mode-covering behaviors induced by the reverse and forward KL divergences in Figure 5, following Le (2017).

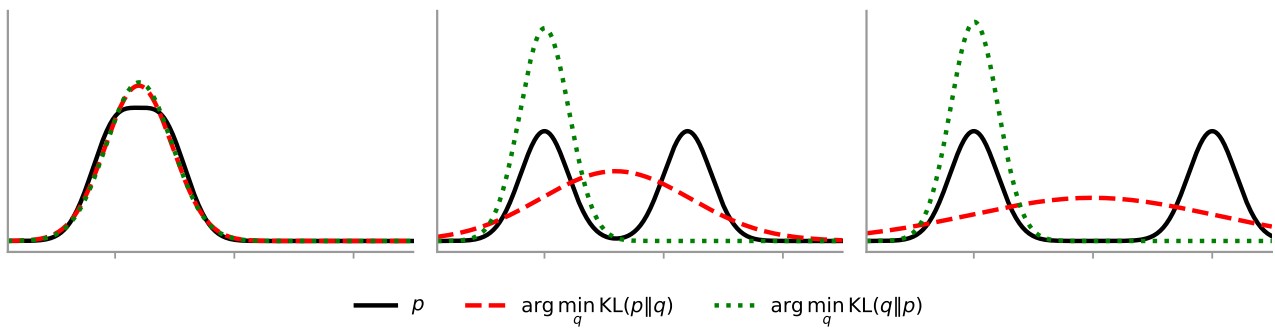

*Figure 5.* **Mode-seeking vs. mode-covering behavior of KL divergences.** Minimizing the reverse-KL divergence encourages the model distribution to concentrate on high-probability modes of the target distribution (mode-seeking), potentially ignoring other valid modes. In contrast, minimizing the forward-KL divergence promotes covering all modes of the target distribution (mode-covering), often at the cost of assigning probability mass to low-density or irrelevant regions.

*Table 5.* **Mean gradient norm across training stages.** We report the mean gradient norm during the Early, Mid, and Late stages of training. GRPO w/o KL consistently exhibits substantially smaller gradient norms throughout training, despite achieving higher training rewards. This suggests rapid convergence toward a narrow reward-optimized policy, after which policy updates quickly saturate. In contrast, KL-regularized variants maintain larger gradient magnitudes, indicating continued non-trivial policy updates and more stable exploration dynamics. These trends are consistent with the reward–performance mismatch observed in Figure 2 and Table 1.

| Algorithm | Early | Mid | Late |
|---|---|---|---|
| GRPO | **0.056** | **0.030** | 0.053 |
| GRPO + Branch | 0.040 | 0.029 | **0.054** |
| GRPO w/o KL | 0.018 | 0.010 | 0.023 |

## B. Training Dynamics Behind Reward Overfitting

As shown in Table 5, GRPO w/o KL exhibits consistently smaller gradient norms throughout training. Despite achieving higher training rewards, its final evaluation performance remains substantially worse, suggesting that the policy rapidly collapses toward a narrow reward-optimized region where learning dynamics quickly saturate. Once such collapse occurs, subsequent updates become small and less effective, limiting further policy improvement and reducing exploration diversity.

In contrast, KL-regularized variants maintain significantly larger gradient magnitudes across all training stages, indicating continued non-trivial policy updates and more stable exploration behavior. This observation is consistent with the reward–performance mismatch observed in Figure 2 and Table 1, supporting the role of reverse-KL regularization in preventing reward hacking behavior.

## C. Toy Example Illustrating the Sufficient Condition in Theorem 4.3

We present a minimal illustrative example demonstrating that the proposed branch-level guide function is *compatible* with the sufficient condition in Theorem 4.3. This example serves purely as a constructive sanity check for the theory, and is not intended to model realistic LLM reasoning dynamics.

**Setup.** Consider a fixed input $x$ with two reward-valid completions $\mathcal{Y} = \{y^1, y^\star\}$: a frequently occurring solution $y^1 = (y_1^1, y_2^1)$ and a rare but valid solution $y^\star = (y_1^\star, y_2^\star)$. Under the reference policy $\pi_{\text{ref}}$, the reasoning path $x \to y_1^1 \to y_2^1$ is traversed with high probability and exhibits low uncertainty, yielding token-level entropies $\mathcal{H}_1^1 = 0.1$ and $\mathcal{H}_2^1 = 0.2$. In contrast, the initial step $x \to y_1^\star$ is substantially less likely and corresponds to a high-entropy branching decision, while the continuation $y_1^\star \to y_2^\star$ is necessary and therefore low-entropy, with $\mathcal{H}_1^\star = 5.0$ and $\mathcal{H}_2^\star = 0.3$.

Let the empirical support threshold be $\epsilon = 10^{-4}$. Assume the reference policy assigns

$$\pi_{\text{ref}}(y^1 \mid x) = 1 - 10^{-6}, \quad \pi_{\text{ref}}(y^\star \mid x) = 10^{-6}. \tag{33}$$

Under this distribution, the rare solution $y^\star$ lies outside the empirical support:

$$y^\star \notin \text{supp}_\epsilon(\pi_{\text{ref}}(\cdot \mid x)).$$

**Branch-Level Guide Construction.** Applying the branch-level guide function defined in Eq. (11) with parameters $(\tau, \gamma) = (1.0, 30)$ yields

$$q(x, y^1) = 1 \times 1 = 1, \quad q(x, y^\star) = \big(1 + 30\,(5.0 - 1.0)\big) \times 1 = 121. \tag{34}$$

Here, amplification is triggered only at the high-entropy branching token, while low-entropy continuation steps remain unaffected.

**Verification of the Sufficient Condition.** Under reverse-KL regularization, relative probabilities among reward-valid solutions are preserved up to a normalization constant. The expected guide value under $\pi_{\text{RLVR}}$ is therefore

$$\mathbb{E}_{y \sim \pi_{\text{RLVR}}}[q(x, y)] = (1 - 10^{-6}) \cdot 1 + 10^{-6} \cdot 121 \approx 1.00012. \tag{35}$$

The sufficient condition in Theorem 4.3 requires

$$q(x, y^\star) > \frac{\epsilon}{\pi_{\text{RLVR}}(y^\star \mid x)} \mathbb{E}_{y \sim \pi_{\text{RLVR}}}[q(x, y)]. \tag{36}$$

Substituting the above values gives

$$\frac{10^{-4}}{10^{-6}} \cdot 1.00012 \approx 100.01 < q(x, y^\star) = 121, \tag{37}$$

and the condition is satisfied.

**Interpretation.** By Theorem 4.3, the stationary distribution induced by SAGE assigns $\pi_{\text{SAGE}}(y^\star \mid x) > \epsilon$, implying that the previously underrepresented but reward-valid solution $y^\star$ enters the empirical support. This example illustrates that entropy-thresholded branch-level amplification can, in principle, satisfy the theoretical sufficient condition without access to explicit reward information. In practical settings, this effect compounds multiplicatively across multiple branching points, such that relatively modest values of $\gamma$ are found sufficient.

## D. Hyperparameter Ablation on Branch-Level Exploration

Table 6 presents a grid search over the branch-level exploration hyperparameters $\tau \in \{0.8, 1.2, 2.0\}$ and $\gamma \in \{0.1, 0.3, 0.5\}$ for GRPO+BRANCH on AMC23. We report pass@1 and pass@256, averaged over 3 random seeds.

The results reveal a clear trade-off between pass@1 and pass@256, with no single configuration consistently dominating across both metrics. For a fixed threshold, larger ratios tend to degrade pass@1, indicating that overly aggressive branch amplification harms low-sample accuracy. When the ratio is moderate or large, increasing the threshold generally improves pass@256, suggesting that selectively triggering exploration under higher uncertainty promotes coverage. In contrast, with a small ratio, more frequent activation via lower thresholds appears beneficial for high-sample performance.

Overall, $(\tau, \gamma) = (1.2, 0.3)$ achieves the most favorable balance between low-sample accuracy and high-sample coverage. Based on this trade-off, we adopt $(\tau, \gamma) = (1.2, 0.3)$ as the default configuration for all main experiments with GRPO+BRANCH.

*Table 6.* **Hyperparameter Ablation on Branch-Level Exploration.** We report pass@1 and pass@256 averaged over 3 seeds.

| Hyperparameters | | AMC23 | |
|---|---|---|---|
| $\tau$ | $\gamma$ | P@1 | P@256 |
| 0.8 | 0.1 | 0.417 | 0.967 |
| 0.8 | 0.3 | 0.450 | 0.958 |
| 0.8 | 0.5 | 0.400 | 0.950 |
| 1.2 | 0.1 | 0.392 | 0.967 |
| 1.2 | 0.3 | 0.433 | 0.967 |
| 1.2 | 0.5 | 0.375 | 0.958 |
| 2.0 | 0.1 | 0.450 | 0.917 |
| 2.0 | 0.3 | 0.400 | 0.983 |
| 2.0 | 0.5 | 0.408 | 0.975 |

## E. Training Details

### E.1. Framework

All training is done using the Unsloth framework (Daniel Han & team, 2023) with vLLM (Kwon et al., 2023) for efficient inference and sampling.

### E.2. Supervised Fine-Tuning

For non-instruction-tuned models, we apply a brief supervised fine-tuning (SFT) stage to calibrate the model's output format and reasoning structure. The training data are restricted to samples whose prompt length does not exceed half of the maximum sequence length. Table 7 lists all hyperparameters used.

*Table 7.* **Hyperparameters for supervised fine-tuning.**

| Parameter | Value |
|---|---|
| Base model | `Qwen2.5-Math-7B-Base` |
| LoRA rank | 8 |
| LoRA target modules | attention & MLP projections |
| Max sequence length | 2,048 |
| Epochs | 2 |
| Batch size | 1 |
| Learning rate | $2 \times 10^{-4}$ |
| Optimizer | AdamW (8-bit) |
| Weight decay | 0.01 |
| LR scheduler | linear |
| Warmup steps | 5 |
| Random seed | 3407 |

## E.3. Reinforcement Learning

Full hyperparameters for the RLVR stage are provided in Table 8. Unless stated otherwise, all experiments in the main text use these settings.

*Table 8.* **Hyperparameters for RLVR training.**

| Parameter | Value |
|---|---|
| Training data | First 2,000 instances of DAPO-Math-17K |
| RL steps | 1,000 |
| Batch size | 8 |
| Rollouts per prompt | 8 |
| Temperature | 1 |
| Min-$p$ | 0.01 |
| Top-$p$ | 1.0 |
| Top-$k$ | $-1$ |
| Learning rate | $5 \times 10^{-6}$ |
| Weight decay | 0.01 |
| LR scheduler | linear |
| Warmup ratio | 0.1 |
| Optimizer | AdamW (8-bit) |
| KL coefficient $\beta$ | 0.05 (when enabled) |
| epsilon | 0.2 |
| epsilon high | 0.28 (if DAPO) |
| Random seed | 3407 |
| Random $\epsilon$ | Cosine schedule $\in [0, 0.1]$ with weight decay 0.9 for a total 8 periods |
| Random $\sigma$ | Cosine schedule $\in [0.05, 0.15]$ with weight decay 0.9 for a total 8 periods |
| SA $\alpha$ | Cosine schedule $\in [0.1, 0.3]$ with weight decay 0.9 for a total 8 periods |
| SA $\sigma$ | Cosine schedule $\in [0.1, 0.25]$ with weight decay 0.9 for a total 8 periods |
| EComp $\gamma$ | 0.3 |
| EComp $\tau$ | 1.2 |

## E.4. Prompt Template

For completeness, we provide the standardized reasoning–and–solution format used throughout SFT, RLVR, and evaluation. The model is instructed with the following system prompt:

```
You are given a problem.
Think about the problem and provide your working out.
Place it between <start_working_out> and <end_working_out>.
Then, provide your solution between <SOLUTION> and </SOLUTION>.
```

The model's generated output therefore follows the format:

```
<start_working_out>
  (model reasoning)
<end_working_out>

<SOLUTION>
  (final numeric answer)
</SOLUTION>
```

The chat template used by the tokenizer follows the Unsloth implementation and is kept fixed across all training stages.

### E.5. Approximating KL Divergence

We approximate $\tilde{D}_{\mathrm{KL}}(\pi_\theta \| q \cdot \pi_{\mathrm{ref}})$ using the second-order unbiased estimator of (Schulman, 2020):

$$\tilde{D}_{\mathrm{KL}}(\pi_\theta \| q \cdot \pi_{\mathrm{ref}}) \approx \frac{q \cdot \pi_{\mathrm{ref}}}{\pi_\theta} - \log \frac{q \cdot \pi_{\mathrm{ref}}}{\pi_\theta} - 1, \tag{38}$$

which is stable and widely used in open-source RLVR frameworks (von Werra et al., 2020; Sheng et al., 2024; Daniel Han & team, 2023).

## F. Evaluation Details

During evaluation, all hyperparameters are kept identical to those used during training, except for the temperature, which is set to 0.3.

## G. Quantitative Reasoning Pattern Analysis

**Experiment setting.** We sampled 16 solutions per problem from the AIME dataset using `Qwen2.5-Math-7B-Base` under three configurations: baseline, GRPO, and GRPO+BRANCH, with a maximum sequence length of 4,096 tokens. Only solutions producing correct final answers were included in the analysis.

**Pattern detection.** We analyzed three representative reasoning patterns: *constraint setup*, *structural reasoning*, and *proof by contradiction*, identified using lightweight keyword-based heuristic detectors.

Specifically, *constraint setup* was detected through symbolic variable introductions (e.g., "let $x =$", "denote $a$", "$a, b, c$"); *structural reasoning* through phrases indicating decomposition or structural forms (e.g., "of the form", "decompose into"); and *proof by contradiction* through explicit contradiction-related statements (e.g., "assume for contradiction", "leads to a contradiction").

**Results and discussion.** Table 2 summarizes the frequency of selected reasoning patterns among correct solutions.

Both GRPO and GRPO+BRANCH substantially increase the occurrence of *constraint setup* and *structural reasoning* compared to the baseline, indicating that RLVR promotes more structured and abstraction-oriented reasoning trajectories.

In contrast, *proof by contradiction* remains nearly absent under the baseline and GRPO, but emerges more frequently under GRPO+BRANCH. This suggests that branch-level exploration enables the model to recover logically valid but low-probability reasoning strategies that are typically suppressed by standard RL optimization.

## H. Case Study on Reasoning Behaviors

**Experiment setting.** We sampled 16 solutions from AIME dataset using `Qwen2.5-Math-7B-Base`: basemodel, GRPO, and GRPO+BRANCH with maximum sequence length of 4,096 tokens. While our quantitative results (Table 1, 3, and 4) demonstrate SAGE's improved coverage via higher pass@k, here we examine a case where *all three models produce the correct answer* to illustrate the qualitative differences in their reasoning processes.

**Problem.** A list of positive integers has the following properties:

- The sum of the items in the list is 30.

- The unique mode of the list is 9.

- The median of the list is a positive integer that does not appear in the list itself.

Find the sum of the squares of all the items in the list.

**Solution by the Base Model.**   Okay, let's start solving this problem. The given conditions are:

1. The sum of the items in the list is 30. 2. The unique mode of the list is 9. 3. The median of the list is a positive integer that does not appear in the list.

First, since the mode is 9 and it is unique, the number 9 must appear more frequently than any other number in the list. Let's denote the number of 9s in the list as $n_9$. Since 9 is the mode and unique, all other numbers must appear fewer times than $n_9$.

The median being a positive integer that does not appear in the list implies that the list has an even number of elements. Let's say the list has $2k$ elements. The median is the average of the $k$-th and $(k + 1)$-th elements in the sorted list. Since the median is a positive integer and does not appear in the list, it must be a number that is not 9.

Let's think about the smallest possible list that satisfies these conditions. If the list has two 9s, the sum of the other numbers must be $30 - 18 = 12$. The median would be the average of the two middle numbers.

If the list is $[9, 9, a, b]$, where $a$ and $b$ are the other numbers, then

$$9 + 9 + a + b = 30 \Rightarrow a + b = 12.$$

**Trying $a = 7$ and $b = 5$ gives the list** $[9, 9, 7, 5]$  with median

$$\frac{9 + 7}{2} = 8,$$

which is a positive integer not in the list.

The sum of squares is

$$9^2 + 9^2 + 7^2 + 5^2 = 81 + 81 + 49 + 25 = 236.$$

Therefore, the answer is $\boxed{236}$.

**Solution by the GRPO-Trained Model.**   Okay, let's tackle this problem step by step.

1. The sum of the items is 30. 2. The unique mode is 9. 3. The median is a positive integer that doesn't appear in the list.

Since the median is not in the list, the list must have an even number of elements.

Assuming four elements with two 9s, the remaining sum is 12.

**Trying $5$ and $7$ gives the list** $[5, 7, 9, 9]$**.**

The median is

$$\frac{5 + 7}{2} = 6,$$

which is not in the list.

The sum of squares is

$$25 + 49 + 81 + 81 = 236.$$

Thus the answer is $\boxed{236}$.

**Solution by the SAGE-Trained Model.**   Okay, let's tackle this problem step by step. So, we have a list of positive integers with the following properties: 1. The sum of the items is 30. 2. The unique mode is 9, so 9 appears more frequently than any other number.

Since the median is not in the list, the number of elements must be even.

Trying two 9s gives remaining sum 12.

**Let the list be $[a, b, 9, 9]$ with $a + b = 12$.**

For the median $\frac{b+9}{2}$ to be an integer not in the list, choose $b = 7$ and $a = 5$.

Then the median is

$$\frac{7+9}{2} = 8.$$

The sum of squares is

$$5^2 + 7^2 + 9^2 + 9^2 = 236.$$

Hence, the answer is $\boxed{236}$.

**Analysis.**   Although all three models arrive at the correct numerical answer, their reasoning traces reveal qualitatively different approaches and levels of rigor.

The base model reaches the correct answer through a trial-and-error approach with some logical gaps. After deriving $a + b = 12$, it assumes an ordering $[9, 9, a, b]$ and computes the median as $(9 + a)/2$. It then tries specific values $a = 7$ and $b = 5$, yielding median $(9 + 7)/2 = 8$. However, the model presents the final constructed list as $[5, 7, 9, 9]$ without acknowledging that this represents a different ordering than initially assumed. While the calculation happens to work out correctly for the assumed ordering $[9, 9, 7, 5]$, this inconsistency in presentation suggests the solution was obtained primarily through trying plausible numbers rather than systematic constraint verification.

The GRPO-trained model exhibits a more pronounced substitution-first pattern with an actual computational error. It assumes the candidate list $[5, 7, 9, 9]$ and computes the median as $(5 + 7)/2 = 6$. This is incorrect-for the sorted list $[5, 7, 9, 9]$, the median should be $(7 + 9)/2 = 8$. Despite this verification error, the model arrives at the correct final answer of 236. This trace reveals that GRPO still predominantly follows a "guess-and-justify" branch, where a plausible candidate is assumed first and then checked (incorrectly), rather than deriving the solution through coherent constraint enforcement.

In contrast, the SAGE-trained model demonstrates a more systematic constraint-verification approach. After fixing two 9's and establishing $a + b = 12$, it explicitly parameterizes the list as $[a, b, 9, 9]$ with $(a, b)$ treated as variables to be determined. It then enforces the median constraint systematically: for the median $\frac{b+9}{2}$ to be an integer not in the list, it selects $(a, b) = (5, 7)$, yielding median $\frac{7+9}{2} = 8$. Rather than post hoc justification of a guessed candidate, the solution emerges by progressively constraining the search space and verifying each requirement in a logically consistent manner.

We hypothesize that this difference reflects SAGE's guided anchor shaping: by reshaping the KL anchor, SAGE can amplify low-probability but reward-relevant verification trajectories that are underrepresented in the pre-training distribution. In this example, the dominant pre-training branch corresponds to heuristic numeric substitution, whereas explicit constraint enforcement represents a rarer but more reliable reasoning mode. By increasing the probability mass on such verification branches during training, SAGE increases the likelihood that the model executes a coherent, constraint-driven solution strategy at inference time.

## I. Additional Experiments

**Additional Benchmark.**   To further validate our SAGE, we conducted additional experiments on the Minerva dataset using `Qwen2.5-Math-7B-Base`.

As shown in Table 9, GRPO improves pass@1 compared to the base model, but substantially degrades pass@256, indicating reduced coverage under larger sampling budgets. In contrast, GRPO+Branch achieves the best pass@1 while preserving more of the high-sample performance than standard GRPO. These results further support that SAGE provides a better accuracy-coverage trade-off by improving single-sample accuracy while mitigating the collapse of diverse correct reasoning trajectories.

*Table 9.* **Additional experiments on Minerva with Qwen2.5-Math-7B-Base.** We report pass@1 and pass@256 on the Minerva benchmark (mean ± std over 3 seeds). While GRPO improves pass@1 at the cost of a substantial degradation in pass@256, GRPO+Branch achieves a better accuracy–coverage trade-off, yielding the strongest pass@1 performance while preserving more high-sample performance. Best results are highlighted in **bold**.

| Algorithm | Pass@1 | Pass@256 |
|---|---|---|
| Base Model | 0.135±0.005 | **0.572±0.014** |
| GRPO | 0.170±0.005 | 0.477±0.008 |
| GRPO + Branch | **0.172±0.002** | 0.483±0.005 |

**Full finetuning.** To show that SAGE is agnostic to finetuning methods, we additionally conducted full finetuning experiments with `Qwen2.5-Math-1.5B-Instruct` and `Qwen2.5-3B-Instruct`.

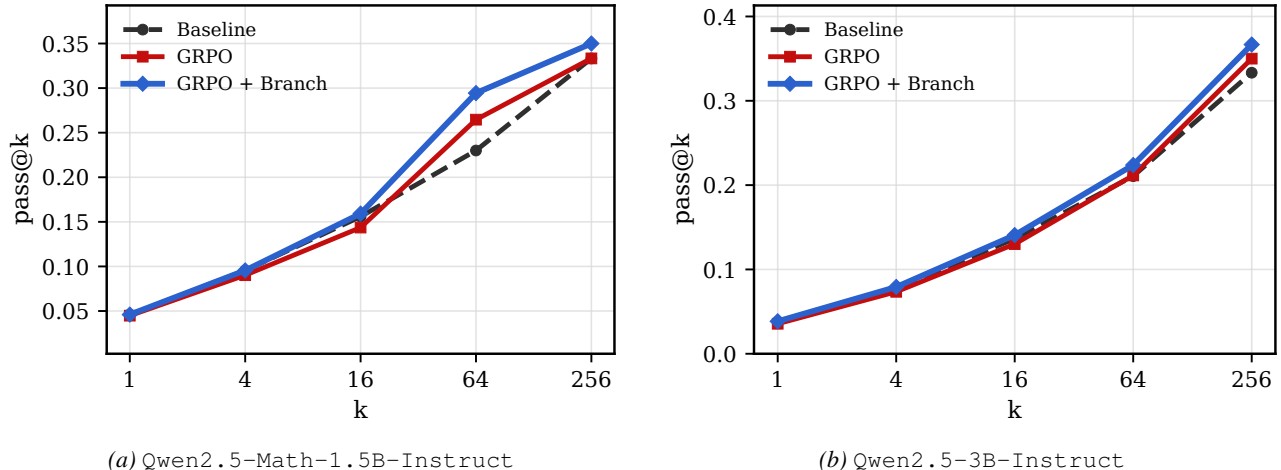

*(a)* `Qwen2.5-Math-1.5B-Instruct`    *(b)* `Qwen2.5-3B-Instruct`

*Figure 6.* **Full finetuning results on smaller-scale models.** We report pass@$k$ on `AIME24/25` estimated from 256 sampled rollouts using the unbiased estimator. While GRPO struggles to improve pass@1 under sparse rewards, SAGE consistently achieves stronger performance across most sampling budgets, particularly in the high-sample regime.

As shown in Figure 6, we observe that smaller-scale models suffer from an extremely sparse learning signal on `DAPO-17k-Processed` dataset, which was originally designed for training substantially larger models. In practice, approximately half of the training samples yield zero advantage, making stable policy improvement difficult under limited computational budgets.

Under this low-signal regime, GRPO provides only marginal gains and often struggles to sufficiently concentrate probability mass toward correct trajectories, despite improving pass@$k$ for larger $k$. In contrast, SAGE consistently improves over standard GRPO across both models, particularly in the high-sample regime. These results suggest that branch-level exploration remains effective even when reward signals are highly sparse, enabling the model to discover and preserve more diverse correct reasoning trajectories during training.

## J. Perplexity Analysis

Building upon prior observations in Yue et al. (2025), we conduct a perplexity-based analysis of model-generated outputs. The underlying hypothesis is that solutions belonging to newly discovered reasoning modes manifest as high-perplexity outliers under the baseline model. The perplexity of an output $Y$ for a question $x$ under model $\pi$ is defined as:

$$\mathrm{PPL}_\pi(Y \mid x) = \exp\left(-\frac{1}{T}\sum_{t=1}^{T} \log \pi(y_t \mid x, y_{<t})\right). \tag{39}$$

**Experimental setting.** To examine mode sharpening effects, we select two geometry problems from AIME—categories known to admit diverse reasoning paths—and analyze their perplexity distributions. We use `Qwen2.5-Math-7B-Base` as the base model and sample 64 responses from each of the following models: the baseline, GRPO, GRPO+RANDOM,

GRPO+TOKEN, and GRPO+BRANCH. All sampled responses are evaluated by computing their perplexity under the baseline model.

**Results and discussion.** Figure 7 presents the resulting perplexity distributions, where $\text{PPL}_{\text{Base}}(Y_r)$ denotes the perplexity of responses generated by model $r$ when evaluated under the baseline model.

Across both problems, the SAGE variants consistently exhibit heavier high-perplexity tails compared to standard RL baselines. Since higher perplexity corresponds to solutions that are less probable under the base model, this pattern indicates that SAGE alleviates mode collapse and preserves access to infrequent and underrepresented reasoning trajectories.

## AIME 2024–I Question No.8

Eight circles of radius 34 are sequentially tangent, and two of the circles are tangent to $\overline{AB}$ and $\overline{BC}$ of $\Delta ABC$ respectively. 2024 circles of radius 1 can be arranged in the same manner. The inradius of triangle $\Delta ABC$ can be expressed as $\frac{m}{n}$ where m and n are relatively prime positive integers. Find m + n.

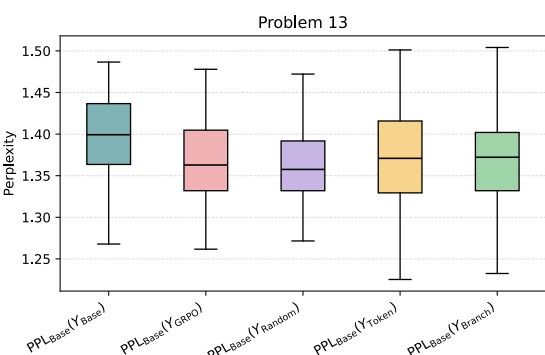

## AIME 2025–I Question No.2

On $\Delta ABC$ points A, D, E and B lie in that order on side $\overline{AB}$ with $\overline{AD} = 4$, $\overline{DE} = 6$ and $\overline{EB} = 8$. Points A, F, G, and C lie in that order on side $\overline{AC}$ with $\overline{AF} = 13$, $\overline{FG} = 52$, and $\overline{GC} = 26$. Let M be the reflection of D through F , and let N be the reflection of G through E. Quadrilateral DEGF has area 288. Find the area of heptagon AFNBCEM.

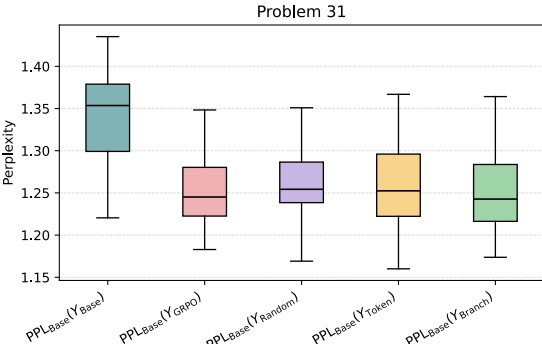

*Figure 7.* **Perplexity analysis on representative AIME problems.** Higher perplexity indicates that a solution is less likely under the base model. **Left:** selected geometry problems. **Right:** perplexity distributions of solutions generated by each model variant. SAGE variants consistently preserve heavier high-perplexity tails, indicating increased resistance to mode collapse.

