# OpenReview forum: "SAGE: Shaping Anchors for Guided Exploration in RLVR of LLMs"
_ICML.cc/2026/Conference — ICML 2026 regular_

### Official Review · Reviewer_M8Fw · 2026-02-16

**Soundness:** 3
**Presentation:** 3
**Significance:** 3
**Originality:** 3
**Overall Recommendation:** 4
**Confidence:** 4

**Summary:**

The paper analyzes why reinforcement learning with verifiable rewards (RLVR) often improves pass@1 accuracy of large language models but yields limited or unstable gains in pass@k. It argues that the common reverse-KL regularization anchors the model too closely to the reference policy, restricting exploration of low-probability yet correct reasoning paths. To address this, the authors propose SAGE (Shaping Anchors for Guided Exploration), a framework that reshapes the KL anchor distribution with lightweight intrinsic signals (e.g., entropy or surprisal) so the model can explore more diverse solutions while maintaining training stability. Experiments on math reasoning benchmarks show that SAGE can simultaneously improve both pass@1 and pass@k, suggesting better coverage of valid reasoning modes without removing the stabilizing KL constraint.

**Compliance With Llm Reviewing Policy:**

Affirmed.

**Final Justification:**

The rebuttal addressed my main concerns, and I have increased both my overall score and my soundness rating to positive.

**Key Questions For Authors:**

See weaknesses. I would consider increasing the score if these issues are clarified or addressed.

**Limitations:**

Yes

**Strengths And Weaknesses:**

Strengths

1. Good Presentation and Clarity. The paper is generally well written and easy to follow, with intuitive figures and well-structured explanations that make the core ideas accessible to the reader.
2. Clear Motivation and Insight. The work identifies an important and practical issue in reinforcement learning with verifiable rewards: the trade-off between pass@1 and pass@k performance induced by reverse-KL anchoring. The analysis provides a clear conceptual explanation of why exploration is restricted under standard reverse-KL regularization.
3. Theoretical Intuition. Both in the main text and the appendices, the paper presents theoretical analysis illustrating how anchor shaping can expand empirical support and promote exploration while maintaining training stability. Although the analysis is not fully rigorous in all practical settings, it offers useful and informative intuition.

Weaknesses

1. Inconsistent pass@k Improvements Across Datasets. Although the paper emphasizes simultaneous gains in both pass@1 and pass@k, the improvements in pass@k are not consistent across all benchmarks. In Table 1, SAGE does not outperform strong baselines such as Forward KL on AIME and AMC23 for pass@k, and the overall improvement mainly comes from MATH-500 and the averaged metric, where the margin is relatively small. This suggests that the benefit may be more dataset-dependent than implied in the main text.
2. Limited Model Scale and Domain Diversity. All experiments are conducted on 7B-scale models and primarily on mathematical reasoning datasets. While this is a reasonable starting point, it remains unclear whether the method generalizes to larger model scales or to other reasoning domains such as commonsense, coding, or multi-step logical reasoning. Broader evaluation would strengthen the practical significance of the work.
3. Limited Visualization of pass@k Behavior.
The paper mainly reports pass@1 and pass@256 (and occasionally pass@64). Presenting a figure of full pass@k curves (accuracy as y axis vs. k as x axis) would provide a more complete view of how exploration and coverage evolve with sampling budget, and would better support the central claim regarding improved trade-offs.
4. Lightweight Reasoning-Pattern Analysis.
The reasoning-pattern analysis relies on keyword-based detectors and is computed only over correct solutions. While illustrative, this proxy may not reliably reflect true reasoning diversity or “mode discovery” without additional qualitative examples, structural trajectory analysis, or human/LLM-judge validation.
5. Unclear Robustness to Hyperparameters.
For the BRANCH variant $q_{\text{Branch}} = 1 + \gamma (H_t - \tau)_+$, the sensitivity to threshold \tau and scaling factor \gamma is not thoroughly characterized across datasets and models. If performance is highly sensitive to these choices, it could reduce the method’s practical robustness and reproducibility.

---

> ### Author Rebuttal · Authors · 2026-03-31
>
> We thank the reviewer for their detailed and insightful feedback.
> We are glad to hear that you found our work (1) presented clearly, (2) has clear motivation and insight and, (3) has theoretical intuition.
> To address your concerns,
> - we add OOD evaluation (Knights & Knaves),
> - report full pass@k,
> - clarify reasoning diversity via multiple analyses,and
> - clarify hyperparameter robustness and usability.
>
> ---
>
> > W1. Inconsistent pass@k vs forward-KL
>
> We thank the reviewer for highlighting this point.
>
> - We clarify that **forward-KL is inherently coverage-oriented**, improving pass@k by spreading probability mass broadly, including off-target regions. In contrast, **SAGE targets a better accuracy–coverage trade-off**, rather than maximizing coverage alone.
> - Empirically, **SAGE consistently improves pass@1 across datasets**, while **maintaining competitive pass@k**, despite not explicitly optimizing for coverage (L361–368, left). We consider this a more desirable regime, as pass@1 directly reflects single-sample reasoning quality.
> - From a theoretical perspective, as discussed in **Appendix A.4**, SAGE **reduces off-target probability mass** compared to forward-KL, leading to more effective allocation toward reward-relevant trajectories.
>
> We have clarified the explanation on forward-KL and this trade-off more explicitly in the revision.
>
> ---
>
> > W2. Limited Model Scale & Domain Generalization
>
> While scaling to larger models is beyond our current computational budget, we agree that demonstrating broader applicability is crucial. To partially address this, we conducted out-of-distribution (OOD) evaluation on the Knights and Knaves logical reasoning benchmark.
>
> | Algorithm   | pass@1 | pass@2 | pass@4 | pass@8 | pass@16 | pass@32 |
> | - | - | - | - | - | - | - |
> | Base Model    | 0.276  | 0.458  | 0.668  | 0.838  | 0.939   | 0.990   |
> | GRPO        | 0.257  | 0.433  | 0.639  | 0.810  | 0.910   | 0.950   |
> | GRPO+Branch | 0.274  | 0.454  | 0.663  | 0.836  | 0.943   | 0.990   |
>
> (pass@ks are computed using unbiased estimator from [1])
>
> As shown in above, we find that **SAGE is more robust under distribution shift**, maintaining strong performance across the entire pass@k curve, whereas GRPO exhibits consistent degradation at higher k.
>
> ---
>
> > W3. Missing pass@k Curves
>
> We thank the reviewer for their valuable suggestion.
>
> To provide a more complete view of exploration–coverage trade-offs, we computed full pass@k curves on AIME, AMC23, and MATH-500, following the standard protocol of [1].
> As the rebuttal does not allow us to include figures, we report representative points from the pass@k curves below.
>
> AIME
> | Algorithm | pass@1 | pass@4 | pass@16 | pass@64 | pass@256 |
> | - | - | - | - | - | - |
> | Base Model | 0.032 | 0.087 | 0.166 | 0.269 | 0.400 |
> | GRPO | 0.058  | 0.159 | 0.256   | 0.321   | 0.417 |
> | GRPO+Branch | 0.064 | 0.160 | 0.259 | 0.348 | 0.450 |
>
> AMC
> | Algorithm | pass@1 | pass@4 | pass@16 | pass@64 | pass@256 |
> | - | - | - | - | - | - |
> | Base Model | 0.356 | 0.534 | 0.777 | 0.900 | 0.950 |
> | GRPO | 0.410 | 0.614 | 0.788 | 0.903 | 0.975 |
> | GRPO+Branch | 0.415 | 0.617 | 0.798 | 0.928 | 1.000 |
>
> MATH-500
> | Algorithm   | pass@1 | pass@4 | pass@16 | pass@64 | pass@256 |
> | - | - | - | - | - | - |
> | Base Model | 0.290 | 0.464 | 0.636 | 0.724 | 0.776 |
> | GRPO | 0.322 | 0.492 | 0.622 | 0.722 | 0.784 |
> | GRPO+Branch | 0.324 | 0.494 | 0.623 | 0.717 | 0.791 |
>
> These results provide a more complete picture of performance across sampling budgets and directly address the reviewer’s concern. In particular, **GRPO+Branch consistently improves pass@1 while maintaining or improving pass@k across a wide range of k**, demonstrating a better accuracy-coverage trade-off rather than gains limited to a single operating point.
> We will include the full pass@k curves in the revision for clearer visualization.
>
> ---
>
> > W4. Weak Reasoning Mode Analysis
>
> We thank the reviewer for their valuable and thoughtful feedback.
>
> We evaluate reasoning diversity using **three complementary analyses**:
> - **Pattern-based analysis** (interpretable signals of strategy differences),
> - **Perplexity-based analysis** (captures distributional shifts and mitigated mode collapse),
> - **Qualitative case studies** (verify meaningful differences in reasoning strategies).
>
> Together, these provide consistent evidence that SAGE recovers **valid but low-probability reasoning modes** that standard RLVR fails to discover.
>
> A more detailed explanation is provided in our response to **Reviewer MMYZ (W5)**.
>
> ---
>
> > W5. Hyperparameter Robustness
>
> We have conducted a detailed hyperparameter ablation study in Appendix C, where we show that performance is not overly sensitive to the choice of $\tau$ and $\gamma$.
>
> For a more detailed discussion and interpretation of these results, please refer to our response to **Reviewer zwME (Q2)**.
>
> #### References
>
> [1] Chen et al., Evaluating Large Language Models Trained on Code

---

> > ### Author Rebuttal · Reviewer_M8Fw · 2026-03-31
> >
> > The rebuttal addresses my main concerns and leads me to revise my score positively.

---

> > > ### Author Response · Authors · 2026-04-01
> > >
> > > Dear Reviewer M8Fw,
> > >
> > > Thank you for your thoughtful consideration, and positive assessment of our work. We truly appreciate your decision to raise the score to 4, and we are glad that the rebuttal **fully addressed your concerns**, including reasoning mode analysis and domain diversity.
> > >
> > > Your feedback has been invaluable in improving the clarity and completeness of the paper. If you have any remaining questions or concerns, please let us know. We are happy to discuss them further.
> > >
> > > Best Regards,
> > >
> > > The Authors

---

### Official Review · Reviewer_YKue · 2026-03-11

**Soundness:** 3
**Presentation:** 3
**Significance:** 3
**Originality:** 3
**Overall Recommendation:** 4
**Confidence:** 4

**Summary:**

The authors propose SAGE, a principled framework that addresses mode collapse in reinforcement learning with verifiable rewards by reshaping the reverse-KL anchor with a guide function derived from intrinsic reasoning signals like entropy and surprisal. This method enables controlled empirical support expansion to discover underexplored reasoning modes, consistently improving both accuracy (pass@1) and solution diversity (pass@256) across challenging mathematical reasoning benchmarks (e.g., AIME, AMC23, MATH-500).

**Compliance With Llm Reviewing Policy:**

Affirmed.

**Key Questions For Authors:**

see weaknesses above.

**Limitations:**

yes

**Strengths And Weaknesses:**

### Strengths

- Because SAGE modifies the anchor distribution rather than the reward structure, it is a plug-and-play modular augmentation that can be integrated into existing RLVR pipelines.
- The introduction of pseudo-KL divergence for unnormalized anchoring distributions provides a new tool for researchers exploring alternative regularization strategies.
- The formal proof of empirical support expansion demonstrates SAGE’s unique capability to recover reward-valid reasoning modes that are theoretically unreachable under standard reinforcement learning objectives.

### Weaknesses

- This paper misses related works and comparison, e.g., RPG: On the Design of KL-Regularized Policy Gradient Algorithms for LLM Reasoning.
- SAGE relies on manually configured heuristic rules for the guide function $q$ , while its applicability to non-verifiable tasks beyond mathematical reasoning, such as open-ended writing, has not been explored.

---

> ### Author Rebuttal · Authors · 2026-03-31
>
> We are grateful to the reviewer for their careful evaluation and constructive suggestions.
> We are pleased that you found our work (1) a simple and plug-and-play augmentation that integrates seamlessly into existing RLVR pipelines, (2) a meaningful contribution through the introduction of pseudo-KL divergence for unnormalized anchoring distributions, and (3) theoretically well-grounded via the formalization of empirical support expansion.
> To address your concerns,
> - we clarify the distinction between our work and prior KL-regularized RL formulations such as RPG
> - we discuss how insights from RPG can be complementary to SAGE
> - and we clarify the scope of our method and its applicability beyond verifiable settings.
>
> ---
>
> > W1. Missing Related Work (RPG)
>
> We sincerely thank the reviewer for pointing out this important and highly relevant work.
> RPG provides a principled analysis of KL-regularized policy optimization by decomposing KL design along axes such as **direction, estimation, and normalization** and studies how to correctly optimize a given objective between fixed distributions. This is a valuable contribution to improving the correctness and stability of KL-based RL.
>
> Our work is **complementary but orthogonal in focus**. While RPG studies how to optimize a given KL objective, we investigate how the choice of anchor distribution itself affects exploration. In particular, we show that even with correctly specified objectives, standard KL regularization (especially reverse-KL) can restrict the empirical support of the policy.
> SAGE addresses this limitation by **reshaping the anchor distribution via a guide function**, enabling controlled support expansion and improved exploration.
>
> Importantly, we believe that RPG’s findings especially on **estimation** can be directly leveraged to further improve SAGE, making the two approaches highly complementary.
> We have included a detailed discussion in the revision.
>
> ---
>
> > W2. Limited to Verifiable Tasks
>
> We sincerely thank the reviewer for this thoughtful and forward-looking comment.
>
> We fully agree that extending to **non-verifiable tasks** (e.g., open-ended writing) is an important and challenging direction. However, reinforcement learning in such settings remains an **open problem**, as defining reliable and stable reward signals is still non-trivial.
>
> Our work is intentionally focused on the **verifiable reward setting** (e.g., mathematical reasoning), where correctness can be clearly evaluated. Within this scope, our goal is to revisit KL-regularized optimization and demonstrate that **anchor shaping** is an effective direction for improving exploration.
>
> We appreciate the reviewer for highlighting this important limitation, and we will clarify the scope of our work more explicitly in the final version.

---

> > ### Author Rebuttal · Reviewer_YKue · 2026-04-01
> >
> > I thank the authors for their rebuttal to address my concerns. I would like to keep my positive score. Good luck.

---

> > > ### Author Response · Authors · 2026-04-02
> > >
> > > Dear Reviewer YKue,
> > >
> > > Thank you for your continued engagement and recognizing that the **rebuttal addressed your concerns**, particularly regarding related work and limitations of verifiable rewards.
> > >
> > > We appreciate your thoughtful feedback throughout the review process, which helped improve the clarity of our paper.
> > >
> > > Please feel free to reach out if any further questions arise.
> > >
> > > Best regards,
> > >
> > > The Authors

---

### Official Review · Reviewer_MMYZ · 2026-03-13

**Soundness:** 2
**Presentation:** 2
**Significance:** 2
**Originality:** 2
**Overall Recommendation:** 4
**Confidence:** 4

**Summary:**

This paper studies exploration in RLVR for reasoning models. The authors argue that reverse-KL regularization is useful for stability, but it also limits exploration. Based on this view, the authors propose a framework called SAGE. The main idea is to reshape the reverse-KL anchor with a guide function q(x, y), instead of removing the KL term or replacing it with forward-KL. The paper studies several guide functions, including random, token-level, and branch-level exploration. Experiments on AIME, AMC23, and MATH-500 show that SAGE can improve both pass@1 and pass@k over GRPO and some other RLVR variants.

**Compliance With Llm Reviewing Policy:**

Affirmed.

**Final Justification:**

My concerns are all addressed and I have raised the score to 4.

**Key Questions For Authors:**

Please see the weaknesses above.

**Limitations:**

Yes.

**Strengths And Weaknesses:**

#### Strengths

- The proposed solution is simple yet effective.
- The paper is easy to follow.
- The paper includes algorithm ablation and model ablation.

#### Weaknesses

- The proposed method is only evaluated on three math benchmarks. The authors are suggested to evaluate on more tasks, such as Minerva Math and OlympiadBench [1]. Also, for AIME and AMC23, the paper only evaluates by running 5 times and reporting the average and std. It would be better to run 16 or 32 times [1] since the evaluation data is quite limited.
- The authors use LoRA in their experiments. However, it is not clear whether the same conclusion still holds when using full fine-tuning. I suggest the authors to evaluate on full fine-tuning as well since [2] has shown that standard LoRA is suboptimal for RLVR.
- The authors only use first 2k instances of DAPO-Math-17K for training. It is unclear whether the same conclusion still holds with a larger training set. I suggest the authors to evaluate on the whole training data as well.
- The branch-level variant mainly uses entropy as the exploration signal. Previous works such as [1, 2] also use this idea for tree-based exploration in RLVR. I think the paper should discuss these works and clearly explain the difference from these methods.
- The reasoning pattern analysis is weak. It only uses lightweight keyword-based heuristics, so I am not sure whether it really shows new reasoning modes.
- The training setting is also limited. The paper only uses the first 2,000 instances of DAPO-Math-17K, and it is unclear whether the same conclusion still holds with a larger training set.

References

[1] Beyond the 80/20 Rule: High-Entropy Minority Tokens Drive Effective Reinforcement Learning for LLM Reasoning.

[2] Evaluating Parameter Efficient Methods for RLVR.

[3] TreeRL: LLM Reinforcement Learning with On-Policy Tree Search.

[4] First Return, Entropy-Eliciting Explore.

---

> ### Author Rebuttal · Authors · 2026-03-31
>
> We thank the reviewer for their thoughtful and valuable comments.
> We are encouraged that you found our work (1) simple yet effective, (2) easy to follow and, (3) has essential ablations on both model and algorithm.
> To address your concerns,
> - we add broader evaluation (Minerva, Knights & Knaves),
> - clarifiy the relation to prior entropy-based methods,
> - provide evidence beyond the 2k+LoRA setting, and
> - explain the reasoning mode analysis.
>
> ---
>
> > W1. Limited Evaluation
>
> We thank the reviewer for the suggestion on broader evaluation and more robust statistics.
>
> To further validate our method SAGE, we conducted additional experiments on the **Minerva dataset** (3 seeds), reporting **pass@1** and **pass@256**:
>
> | Algorithm | pass@1 | pass@256 |
> | - | - | - |
> | Base Model   | 0.135 $\pm$ 0.005 | 0.572 $\pm$ 0.014 |
> | GRPO       | 0.170 $\pm$ 0.005 | 0.477 $\pm$ 0.008 |
> | GRPO+Branch | 0.172 $\pm$ 0.002 | 0.483 $\pm$ 0.005 |
>
> - These results show that SAGE achieves a better Pareto frontier between Pass@1 and Pass@256 compared to standard GRPO, supporting a better accuracy-coverage trade-off.
> - We also evaluate on an **out-of-distribution logical reasoning task** (Knights and Knaves), where SAGE shows improved robustness under distribution shift (details in Reviewer M8Fw (W2)).
> - Regarding broader statistics, due to the limited rebuttal period and the substantial computational cost of our main experiments (5 seeds with pass@256), scaling to 16-32 runs was not feasible. We believe the consistent gains across Minerva and Knights & Knaves provide robust evidence of SAGE's generalizability.
>
> ---
>
> > W2,W3&W6. Dataset Scale Limitation & LoRA(2k vs Full 17k, LoRA vs Full Fine-Tuning)
>
> We agree that evaluation on the full dataset and full fine-tuning would strengthen validation. To demonstrate that SAGE's benefits scale with data, we conducted a larger-scale experiment (5k data, 500 steps, LoRA) evaluated on AIME:
>
> | Algorithm   | pass@1 | pass@4 | pass@16 | pass@64 | pass@256 |
> | - | - | - | - | - | - |
> | Base Model    | 0.032  | 0.087  | 0.166   | 0.269   | 0.400    |
> | GRPO        | 0.045  | 0.113  | 0.210   | 0.319   | 0.383    |
> | GRPO+Branch | 0.045  | 0.114  | 0.217   | 0.348   | 0.467    |
>
> **SAGE consistently outperforms GRPO as the dataset scale increases**, suggesting that the gains are not specific to the 2k dataset setting.
>
> Regarding LoRA: SAGE operates via anchor shaping in the objective, making its mechanism independent of the parameterization method. Following existing evidence [1] that LoRA can be competitive in RL for LLMs, we focused on establishing the algorithmic superiority of SAGE. We will conduct scaling our experiments to the full dataset and full fine-tuning if budget allows.
>
> ---
>
> > W4. Entropy Exploration vs Prior works(80/20 rule, Tree-RL)
>
> We thank the reviewer for highlighting relevant prior works [2,3].
>
> While both approaches use entropy as an exploration signal, prior methods introduce alternative optimization structures (e.g., tree search), requiring significant changes to the training pipeline. In contrast, SAGE is a lightweight, modular mechanism that enhances exploration purely through anchor shaping within standard reverse-KL frameworks.
>
> Therefore, **SAGE is orthogonal to these approaches and can be seamlessly combined with them**, as all methods share the same underlying reverse-KL regularization. We have clarified these distinction in Section 2 of the revison.
>
> ---
>
> > W5. Weak Reasoning Mode Analysis
>
> We appreciate the feedback on reasoning diversity.
>
> We define a *reasoning mode* as a class of solution trajectories differing in underlying problem-solving strategy (e.g., algebraic vs. constructive reasoning).
>
> We already analyzed reasoning modes using three complementary approaches in the paper:
> - **Pattern-based analysis (Table 2):**
>   Identifies interpretable strategy differences; SAGE more frequently exhibits *proof-by-contradiction*, indicating recovery of valid but low-probability trajectories (L345–353(right)).
> - **Perplexity-based analysis (Appendix H):**
>   Captures distributional shifts; SAGE shows heavier high-perplexity tails, suggesting reduced mode collapse and improved coverage of rare reasoning paths (L1030–1032).
> - **Qualitative case studies (Appendix F):**
>   Provide semantic validation; SAGE favors constraint-driven reasoning (low-probability but valid), while baselines rely more on substitution-based approaches (L988–1011).
>
> Together, these analyses consistently support improved reasoning diversity.
>
> #### References
>
> [1] Schulman et al., LoRA Without Regret
>
> [2] Wang et al., Beyond the 80/20 Rule: High-Entropy Minority Tokens Drive Effective Reinforcement Learning for LLM Reasoning
>
> [3] Hou et al., TreeRL: LLM Reinforcement Learning with On-Policy Tree Search

---

> > ### Author Rebuttal · Reviewer_MMYZ · 2026-04-03
> >
> > Thanks for your detailed reply and additional experimental results. W1, W4, and W5 are all addressed. As for W2, W3, and W6, if the authors are limited by the computational resources, I recommend the authors run **1.5B** models using the **full** training dataset with **full** parameter training. I believe this setup will not require unaffordable GPUs or training time. [1] shows LoRA with carefully tuned hyperparameters can outperform full tuning. However, for many cases, LoRA is still optimal compared to full tuning, especially when lr<1e-5 (as shown in Fig. 6). As shown in Tab. 7 in Appendix D, the authors use lr=5e-6 for RLVR experiments, which corresponds to the case that LoRA is suboptimal to full tuning in [1]. If the authors could provide the results before April 7 AoE (Author-Reviewer Discussion DDL), I will raise my score accordingly.
> >
> > [1] Schulman et al., LoRA Without Regret

---

> > > ### Author Response · Authors · 2026-04-08
> > >
> > > Dear Reviewer MMYZ,
> > >
> > > Thank you for your thoughtful suggestion regarding **full-dataset full fine-tuning**.
> > > We conducted additional experiments using full-dataset full fine-tuning on both
> > > **Qwen2.5-Math-1.5B-Instruct** and **Qwen2.5-3B-Instruct**.
> > >
> > > We observed that DAPO-17K, originally designed for training much larger models (e.g., 32B DeepSeek-distilled models), is **too challenging for smaller-scale models (1.5B–3B)**. As a result, the effective learning signal becomes highly sparse—**approximately half of the training samples yield zero advantage**.
> > > This issue is further exacerbated compared to recent works based on the 1.5B model (e.g., ProRL [1]), which rely on large batch sizes (e.g, 256 with 16 rollouts) and extensive training time (e.g., 16k GPU hours), making it difficult to reproduce similar training dynamics under our computational constraints.
> > >
> > > Despite these challenges, we obtained the following results:
> > >
> > > Qwen2.5-Math-1.5B-Instruct
> > > | Algorithm   | pass@1 | pass@4 | pass@16 | pass@64 | pass@256 |
> > > | - | - | - | - | - | - |
> > > | Baseline    | 0.0458 | 0.0954 | 0.1559  | 0.2300  | 0.3333   |
> > > | GRPO        | 0.0448 | 0.0902 | 0.1436  | 0.2646  | 0.3333   |
> > > | GRPO+Branch | 0.0460 | 0.0955 | 0.1592  | 0.2944  | 0.3500   |
> > >
> > >
> > > Qwen2.5-3B-Instruct
> > > | Algorithm   | pass@1 | pass@4 | pass@16 | pass@64 | pass@256 |
> > > | - | - | - | - | - | - |
> > > | Baseline    | 0.0376 | 0.0773 | 0.1365  | 0.2107  | 0.3333   |
> > > | GRPO        | 0.0356 | 0.0735 | 0.1301  | 0.2115  | 0.3500   |
> > > | GRPO+Branch | 0.0385 | 0.0792 | 0.1407  | 0.2238  | 0.3667   |
> > >
> > > While pass@$k$ with larger $k$s consistently improves, the gains do not translate strongly into pass@1, indicating that the **model struggles to sufficiently concentrate probability mass due to sparse rewards**.
> > > Importantly, even under this low-signal regime, our method (SAGE) demonstrates **consistent improvements over GRPO**, suggesting that it enables more effective exploration.
> > > We appreciate your insightful suggestion and agree that full-dataset full fine-tuning is an important evaluation setting. We will continue refining these experiments and aim to include improved results in the revision.
> > >
> > > Best regards,
> > >
> > > The Authors
> > >
> > > #### References
> > >
> > > [1] Liu et al., ProRL: Prolonged Reinforcement Learning Expands Reasoning Boundaries in Large Language Models

---

### Official Review · Reviewer_zwME · 2026-03-16

**Soundness:** 3
**Presentation:** 3
**Significance:** 3
**Originality:** 3
**Overall Recommendation:** 4
**Confidence:** 4

**Summary:**

This paper presents SAGE (Shaping Anchors for Guided Exploration), a framework designed to address the pervasive issue of mode collapse and limited exploration in Reinforcement Learning with Verifiable Rewards (RLVR) for Large Language Models. Standard RLVR typically relies on a reverse-KL divergence constraint to anchor the learned policy $\pi_{\theta}$ to a reference model $\pi_{ref}$ for training stability; however, this multiplicative update mechanism heavily amplifies existing high-frequency reasoning paths while suppressing valid but rare trajectories. To overcome this without resorting to unstable KL removal or off-target forward-KL optimization, SAGE actively reshapes the anchor distribution by introducing a guide function $q(x,y)$, altering the target to an unnormalized $q \cdot \pi_{ref}$ and optimizing it via a tractable pseudo-KL divergence. Among various designs, the branch-level exploration strategy proves most effective, utilizing token-level information entropy $\mathcal{H}_t$ as a trigger to dynamically amplify exploration only when the model faces high-uncertainty reasoning forks that exceed a specific threshold $\tau$. Theoretically, the authors prove that this guided anchor shaping allows the model to explicitly expand its empirical support beyond the strict limits of the reference policy, enabling true mode discovery. Extensive experiments demonstrate that integrating SAGE into algorithms like GRPO, BNPO, and DAPO consistently yields simultaneous improvements in both accuracy (pass@1) and diversity (pass@k) across challenging mathematical benchmarks like AIME, AMC23, and MATH-500, successfully unlocking rare, high-level logical patterns such as proof by contradiction.

**Compliance With Llm Reviewing Policy:**

Affirmed.

**Key Questions For Authors:**

- Could you provide more detailed training dynamics for the "GRPO w/o KL" baseline? Specifically, plots showing the evolution of gradient norms or the raw KL divergence from the reference policy over training steps would strongly reinforce your premise regarding instability.
- Given that the branch-level exploration relies on a heuristic $\tau$ threshold, how sensitive is the empirical support expansion to slight variations in this hyperparameter across different base models?
- The current $q(x,y)$ relies on intrinsic signals. Do you foresee a tractable way to learn a parameterized $q_\phi(x,y)$ during the RL process to satisfy the condition in Theorem 4.3 more rigorously, perhaps by utilizing a lightweight value model to estimate the potential of rare branches?

**Limitations:**

yes

**Strengths And Weaknesses:**

Strengths
- Theoretical Novelty and Elegance: The mathematical reinterpretation of the reverse-KL divergence is highly commendable. Rather than treating the KL penalty as a rigid constraint to be relaxed or discarded, treating it as a moldable anchor via the guide function $q(x,y)$ provides a principled and creative mechanism for guided exploration.
- Solid Empirical Validation: The experimental design is comprehensive and well-executed. The authors demonstrate that SAGE is not tied to a single algorithm but serves as a modular, plug-and-play augmentation that consistently improves the accuracy-coverage trade-off across heterogeneous RLVR frameworks (GRPO, BNPO, DAPO).
- Meaningful Behavioral Analysis: The quantitative reasoning pattern analysis (e.g., the increased frequency of "Proof by Contradiction") effectively translates the abstract metric improvements (pass@k) into tangible evidence of behavioral shifts and mode discovery within the model's reasoning trajectories.

Weaknesses
- Theory-Practice Gap: The theoretical framework, specifically Theorem 4.3, is fundamentally an existence proof. It establishes that an optimal guide function can expand the empirical support. However, the practical instantiation proposed—an entropy-based thresholding mechanism for branch-level exploration—is entirely heuristic. The paper lacks a rigorous theoretical bridge connecting the sufficient condition outlined in Eq. (8) to the empirical entropy heuristic, weakening the impact of the theoretical claims.
- Fundamental Support Boundary Limitations: As acknowledged in the appendix, SAGE is inherently constrained by the theoretical support of the reference policy $\pi_{ref}$. It redistributes probability mass among trajectories that already possess non-zero likelihood but cannot achieve true out-of-distribution mode discovery. If a valid reasoning trajectory is entirely absent from the pre-training distribution, this method cannot recover it.
- Insufficient Evidence for the Core "Instability" Premise: A central motivation of the paper is that removing the KL constraint entirely leads to severe optimization instability or reward hacking. While Figure 2 and Table 1 show that dropping the KL term degrades pass rates despite higher training rewards, the paper lacks granular, compelling evidence of true instability. To fully justify the necessity of the SAGE framework over simpler unconstrained methods, the authors should provide a deeper ablation demonstrating the underlying optimization failures when the KL term is removed (e.g., tracking gradient norms, policy KL explosion, or severe model overconfidence during the RL updates).

---

> ### Author Rebuttal · Authors · 2026-03-31
>
> We thank the reviewer for their detailed and insightful feedback.
> We are glad that you found our work (1) theoretically novel and elegant, (2) empirically solid, and (3) behaviorally meaningful.
> To address your concerns,
> - we clarify our theory as an existence result and discuss the gap to the current heuristic, while outlining directions for learning a parameterized guide
> - we position SAGE within the fundamental support limitation of RL and highlight its complementary role with SFT/distillation
> - we refine our motivation by distinguishing reward–performance mismatch from optimization instability
>
> ---
>
> > W1&Q3. Theory-Practice Gap
>
> We appreciate the reviewer for their insightful comment and suggestion.
> Our work introduces **anchor shaping**, which explores **modifying the KL anchor—rather than removing KL—as a new direction for improving exploration.** We intentionally adopt a simple entropy-based heuristic (Branch) to demonstrate that even lightweight guide functions can yield meaningful gains.
>
> While Theorem 4.3 provides the theoretical foundation by establishing the existence of an optimal guide, our current mechanism is designed as a practical, lightweight approximation. By leveraging intrinsic signals as a minimal proxy, we can construct simple, highly effective guide functions (e.g., random, token, and branch in Section 4.3).
>
> We particularly find the suggestion of learning a **parameterized guide** $q_\phi(x,y)$ highly promising. For instance, lightweight value or uncertainty models could provide a better approximation of the optimal guide, as clarified in Appendix I.
>
> ---
>
> > W2. Fundamental Support Limitation
>
> We clarify that SAGE focuses on **maximizing utilization of the existing model support** rather than out-of-distribution discovery.
> - While expanding support typically requires SFT or distillation [1], **SAGE instead redistributes probability mass toward high-reward regions within that support.**
> - In practice, RL often fails to identify optimal modes even when they exist (Figure 1); SAGE addresses this via anchor shaping.
> - Thus, **SAGE is complementary to support-expanding methods**, ensuring newly introduced capabilities are effectively exploited.
>
> We will clarify this positioning and explicitly discuss this limitation in the revision.
>
> ---
>
> >  W3&Q1. KL Removal: Reward–Overfitting
>
> We would like to first clarify that **our paper does not claim optimization instability** (e.g., exploding gradients or divergence) when removing the KL constraint. Instead, our motivation is centered on a different phenomenon.
>
> - As shown in Figure 2 and Table 1, removing the reverse-KL penalty leads to a clear **reward–overfitting**: the training reward continues to increase, while the actual task performance (e.g., pass rates) degrades. Our goal is to highlight this **reward-hacking behavior**, rather than optimization instability in the classical sense, as explained in L356-361(left).
> - We agree this distinction was not sufficiently emphasized and will revise to clarify that the issue is not optimization instability but **generalization degradation due to misaligned reward optimization** in the revision.
> - To support our claim, we report mean gradient norm in Early/Mid/Late term during training.
>
> | Algorhithm | Early | Mid | Late |
> | - | - | -| - |
> | GRPO           | 0.056  | 0.030  | 0.053 |
> | GRPO + Branch  | 0.040  | 0.029  | 0.054 |
> | GRPO w/o KL    | 0.018  | 0.010  | 0.023 |
>
> - GRPO w/o KL exhibits consistently lower gradient norms, indicating premature convergence to a suboptimal, reward-overfitted policy. In contrast, KL-regularized variants maintain higher gradients, reflecting continued meaningful updates, consistent with the reward–performance mismatch in Figure 2 and Table 1.
>
> ---
>
> > Q2. Hyperparameter Sensitivity
>
> We reported an ablation study on hyperparameters in Appendix C.
>
> The results show that **performance is not overly sensitive to the choice of $\tau$ and $\gamma$**. In particular, pass@1 consistently peaks at a moderate value of $\gamma$ (around 0.3) across different $\tau$, indicating stable behavior. While pass@256 exhibits some variation with respect to $\tau$, the overall trends remain consistent and do not require fine-grained tuning to achieve strong performance.
>
> Importantly, the observed patterns reflect an inherent trade-off between exploitation (pass@1) and coverage (pass@256), rather than instability. This suggests that the hyperparameters control interpretable aspects of the exploration–exploitation balance, and reasonable default values (e.g., $\gamma=0.3$) work well across settings.
>
> #### References
>
> [1] Yue et al., Does Reinforcement Learning Really Incentivize Reasoning Capacity in LLMs Beyond the Base Model?

---

> > ### Author Rebuttal · Reviewer_zwME · 2026-04-03
> >
> > Thank you for your detailed response and added results. I will keep my score as is.

---

> > > ### Author Response · Authors · 2026-04-04
> > >
> > > Dear Reviewer zwME,
> > >
> > > Thank you for your continued engagement. We are pleased that our **rebuttal fully clarified the concerns** around the w/o KL baseline, particularly its reward hacking behavior.
> > >
> > > We appreciate your careful reading and helpful feedback throughout the review process.
> > >
> > > Best regards,
> > >
> > > The Authors

---

### Decision · Program_Chairs · 2026-04-30

**Decision:**

Accept (regular)

**Comment:**

The paper proposes SAGE to improve exploration in RLVR. The key idea is to introduce a guide function that biases exploration toward underrepresented but potentially reward-valid reasoning trajectories while retaining the stabilizing role of reverse-KL regularization. The paper provides a pseudo-KL formulation for unnormalized anchors, a theoretical argument for empirical support expansion, and empirical results showing that the branch-based variant improves the accuracy–coverage trade-off across GRPO, BNPO, and DAPO on AIME, AMC23, and MATH-500.

Reviewers agreed on several strengths. They viewed the central idea as novel and well motivated;. They also found the empirical study reasonably broad within the chosen setting. Some reviewers also appreciates the behavioral analyses. It hs aslo a consensus that the idea is simple and effective.

The main weaknesses were also consistent across reviews. First, the theory-to-practice connection remains incomplete: the main theorem is an existence result, while the practical guide function is heuristic and entropy-based. Second, the method is fundamentally limited to redistributing probability mass within the support of the reference policy, so it does not address genuinely novel reasoning outside that support. Third, the experimental scope was initially narrow: only three math benchmarks, 7B-scale models, LoRA-based training, and training on only the first 2k examples. Reviewers also asked for clearer comparison to related entropy-based exploration methods and RPG, fuller pass@k characterization, and stronger evidence that the reported reasoning-mode shifts go beyond keyword heuristics.

The rebuttal addressed many of of these concerns. However, the core theoretical gap remains only partially addressed, and the evidence beyond the original 7B/LoRA setting is still limited. Overall, the rebuttal substantially improves  the paper.